# A synergy-based hand control is encoded in human motor cortical areas

Andrea Leo[1,2], Giacomo Handjaras[1], Matteo Bianchi[2,3], Hamal Marino[2], Marco Gabiccini[2,3,4], Andrea Guidi[2], Enzo Pasquale Scilingo[2,5], Pietro Pietrini[1,2,6,7], Antonio Bicchi[2,3], Marco Santello[8], Emiliano Ricciardi[1,2]*

[1]Laboratory of Clinical Biochemistry and Molecular Biology, University of Pisa, Pisa, Italy; [2]Research Center 'E. Piaggio', University of Pisa, Pisa, Italy; [3]Advanced Robotics Department, Istituto Italiano di Tecnologia, Genova, Italy; [4]Department of Civil and Industrial Engineering, University of Pisa, Pisa, Italy; [5]Department of Information Engineering, University of Pisa, Pisa, Italy; [6]Clinical Psychology Branch, Pisa University Hospital, Pisa, Italy; [7]IMT School for Advanced Studies Lucca, Lucca, Italy; [8]School of Biological and Health Systems Engineering, Arizona State University, Tempe, United States

**Abstract** How the human brain controls hand movements to carry out different tasks is still debated. The concept of *synergy* has been proposed to indicate functional modules that may simplify the control of hand postures by simultaneously recruiting sets of muscles and joints. However, whether and to what extent synergic hand postures are encoded as such at a cortical level remains unknown. Here, we combined kinematic, electromyography, and brain activity measures obtained by functional magnetic resonance imaging while subjects performed a variety of movements towards virtual objects. Hand postural information, encoded through kinematic synergies, were represented in cortical areas devoted to hand motor control and successfully discriminated individual grasping movements, significantly outperforming alternative somatotopic or muscle-based models. Importantly, hand postural synergies were predicted by neural activation patterns within primary motor cortex. These findings support a novel cortical organization for hand movement control and open potential applications for brain-computer interfaces and neuroprostheses.

*For correspondence: emiliano. ricciardi@bioclinica.unipi.it

**Competing interests:** The authors declare that no competing interests exist.

## Introduction

Unique among primates, the human hand is capable of performing a strikingly wide range of movements, characterized by a high degree of adaptability and dexterity that enables complex interactions with the environment. This is exemplified by the hand's ability to mold to objects and tools by combining motion and force in the individual digits so to reach a variety of hand postures. The multiple ways in which the hand can perform a given goal-directed movement arise from anatomical, functional, and kinematic redundancies, i.e., a large number of degrees of freedom (DoFs) (**Bernstein, 1967**). Such an organization results highly advantageous from an operational perspective, as redundant DoFs enable the hand to flexibly adapt to different task demands, or to switch among multiple postural configurations, while maintaining grasp stability (**Bernstein, 1967**; **Santello et al., 2013**). At the same time, this organization raises the question about how the central nervous system deals with these redundancies and selects a set of DoFs to accomplish a specific motor task (**Latash et al., 2007**). While some models propose the notion of "freezing" of redundant DoFs (**Vereijken et al., 1992**) or the implementation of optimization strategies (**Flash and Hogan, 1985**;

**eLife digest** The human hand can perform an enormous range of movements with great dexterity. Some common everyday actions, such as grasping a coffee cup, involve the coordinated movement of all four fingers and thumb. Others, such as typing, rely on the ability of individual fingers to move relatively independently of one another.

This flexibility is possible in part because of the complex anatomy of the hand, with its 27 bones and their connecting joints and muscles. But with this complexity comes a huge number of possibilities. Any movement-related task – such as picking up a cup – can be achieved via many different combinations of muscle contractions and joint positions. So how does the brain decide which muscles and joints to use?

One theory is that the brain simplifies this problem by encoding particularly useful patterns of joint movements as distinct units or "synergies". A given task can then be performed by selecting from a small number of synergies, avoiding the need to choose between huge numbers of options every time movement is required.

Leo et al. now provide the first direct evidence for the encoding of synergies by the human brain. Volunteers lying inside a brain scanner reached towards virtual objects – from tennis rackets to toothpicks – while activity was recorded from the area of the brain that controls hand movements. As predicted, the scans showed specific and reproducible patterns of activity. Analysing these patterns revealed that each corresponded to a particular combination of joint positions. These activity patterns, or synergies, could even be 'decoded' to work out which type of movement a volunteer had just performed.

Future experiments should examine how the brain combines synergies with sensory feedback to allow movements to be adjusted as they occur. Such findings could help to develop brain-computer interfaces and systems for controlling the movement of artificial limbs.

*Todorov and Jordan, 2002*; *Todorov, 2004*), further studies have favored an alternative solution based on linear dimensionality reduction strategies or *motor synergies* (*Latash, 2010*).

From a theoretical perspective, synergies represent functional sensorimotor modules that result from the combination of elementary variables and behave as single functional units (*Turvey, 2007*; *Latash, 2010*). From an experimental viewpoint, synergy-based models have been applied with success to electrophysiological and kinematic data acquired in frogs (*d'Avella and Lacquaniti, 2003*; *Cheung et al., 2005*), monkeys (*Overduin et al., 2012*) and humans (*Bizzi et al., 2008*).

With regard to hand control in humans, synergies have been defined at different levels. *Kinematic* synergies correspond to covariation patterns in finger joint angles and are quantified through kinematic recordings (*Santello et al., 1998*; *Gabiccini et al., 2013*; *Tessitore et al., 2013*). *Muscle* synergies represent covariation patterns in finger muscle activations and are typically extracted from electromyography (EMG) signals (*Weiss and Flanders, 2004*; *d'Avella and Lacquaniti, 2013*).

The first quantitative description of kinematic hand synergies was obtained by analyzing hand postures used by subjects for grasping imagined objects that varied in size and shape (*Santello et al., 1998*). Three hand postural synergies were identified through a principal component analysis (PCA) that accounted for a high fraction (>84%) of variance in the kinematic data across all hand postures and characterized hand configurations as linear combinations of finger joints (*Santello et al., 1998*). Notably, other studies achieved similar results using kinematic data acquired during grasping of real, recalled and virtual objects (*Santello et al., 2002*), exploratory procedures (*Thakur et al., 2008*), or during different movements, such as typing (*Soechting and Flanders, 1997*), as well as with EMG signals from finger muscles during hand shaping for grasping or finger spelling (*Weiss and Flanders, 2004*).

Given that final hand postures can be described effectively as the linear combination of a small number of synergies, each one controlling a set of muscles and joints, the question arises whether kinematic or muscular hand synergies merely reflect a behavioral observation, or whether instead a synergy-based framework is grounded in the human brain as a code for the coordination of hand movements. According to the latter hypothesis, motor cortical areas and/or spinal modules may

control the large number of DoFs of the hand through weighted combinations of synergies (*Gentner and Classen, 2006*; *Santello et al., 2013*; *Santello and Lang, 2014*), in a way similar to that demonstrated for other motor acts, such as gait, body posture, and arm movements (*Cheung et al., 2009*). Furthermore, the biomechanical constraints of the hand structure that group several joints in nature (e.g., multi-digit and multi-joint extrinsic finger muscles whose activity would generate coupled motion), are compatible with the synergistic control of hand movements.

Previous brain functional studies in humans are suggestive of a synergistic control of hand movements. For instance, in a functional magnetic resonance imaging (fMRI) study, synergistic/dexterous and non-synergistic hand movements elicited different neural responses in the premotor and parietal network that controls hand posture (*Ehrsson et al., 2002*). Equally, transcranial magnetic stimulation (TMS) induced hand movements encompassed within distinct postural synergies (*Gentner and Classen, 2006*). Despite all the above pieces of information, however, *whether* and *to what extent* the representation of hand movements is encoded at a cortical level in the human brain directly as postural synergies still remains an open question.

Alternative solutions to synergies for hand control have been proposed as well. Above all, classic *somatotopic* theories postulated that distinct clusters of neuronal populations are associated with specific hand muscles, fingers, or finger movements (*Penfield and Boldrey, 1937*; *Penfield and Rasmussen, 1950*; *Woolsey et al., 1952*). However, whereas a coarse arrangement of body regions (e. g., hand, mouth, or face) has been shown within primary motor areas, the intrinsic topographic organization within limb-specific clusters remains controversial. In hand motor area, neurons controlling single fingers are organized in distributed, overlapping cortical patches without any detectable segregation (*Penfield and Boldrey, 1937*; *Schieber, 1991*, *Schieber, 2001*). In addition, it has been recently shown that fMRI neural activation patterns for individual digits in sensorimotor cortex are not somatotopically organized and their spatial arrangement is highly variable, while their representational structure (i.e., the pattern of distances between digit-specific activations) is invariant across individuals (*Ejaz et al., 2015*).

The present study was designed to determine whether and to what extent synergistic information for hand postural control is encoded as such at a neural level in the human brain cortical regions.

An identical experimental paradigm was performed in two distinct sessions to acquire kinematic and electromyographic (EMG) data while participants performed grasp-to-use movements towards virtual objects. Kinematic data were analyzed according to a *kinematic synergy* model and an *individual-digit* model, based on the independent representation of each digit (*Kirsch et al., 2014*), while EMG data were analyzed according to a *muscle synergy* model to obtain independent descriptions of each final hand posture. In a separate fMRI session, brain activity was measured in the same participants during an identical motor task.

Hence, encoding techniques (*Mitchell et al., 2008*) were applied to brain functional data to compare the synergy-based model with the alternative somatotopic and muscular models on the basis of their abilities to predict neural responses. Finally, to assess the specificity of the findings, we applied a decoding procedure to the fMRI data to predict hand postures based on patterns of fMRI activity.

## Results

### Motion capture and EMG sessions: discrimination accuracy of different models on behavioral data

The hand kinematic data, acquired from the motion capture experiment, provided a *kinematic synergy* description, created using PCA on digit joint angles, and an *individual digit* description, i.e., a somatotopic model based on the displacements of single digits, calculated as the average displacement of their joint angles. The EMG data provided a *muscle synergy* description. To obtain comparable descriptions of hand posture, three five-dimensions models were chosen. A validation procedure based on a rank-accuracy measure was performed to assess the extent to which static hand postures could be reliably discriminated by each behavioral model, regardless of its fraction of variance accounted for. All the three models were able to significantly distinguish between individual hand postures (average accuracy ± standard deviation -SD-; chance level: 50%; *kinematic synergy*: 91.1 ± 3.6%; *individual digit*: 85.9 ± 5%; *muscle synergy*: 72 ± 7.7%) (*Supplementary file 1A*). Specifically, the *kinematic synergy* model performed significantly better than both the *individual digit* and

*muscle synergy* models while the *individual digit* model was significantly more informative than the *muscle synergy* model (Wilcoxon signed-rank test, *p*<0.05, Bonferroni-Holm corrected).

## fMRI session: discrimination accuracy of different models in single-subject encoding of hand posture

Three independent encoding procedures (*Mitchell et al., 2008*) were performed on the fMRI data to assess to what extent each model (*kinematic synergy, individual digit* or *muscle synergy*) would predict brain activity. The discrimination accuracy was tested for significance against unique null distributions of accuracies for each participant and model obtained through permutation tests.

Overall, the encoding procedure based on the *kinematic synergy* model was highly successful across all participants (average accuracy ± SD: 71.58 ± 5.52%) and always significantly above chance level (see *Supplementary file 1B* for single subject results). The encoding of the *individual digit* model was successful in five of nine participants only (63.89 ± 6.86%). Finally, the *muscle synergy* model successfully predicted brain activity in six out of eight participants, with an average accuracy that was comparable to the *individual digit* model (63.9 ± 6.5%).

The *kinematic synergy* model outperformed both the *individual digit* and the *muscle synergy* models (Wilcoxon signed-rank test, *p*<0.05, Bonferroni-Holm corrected), whereas no significant difference was found between the *individual digit* and *muscle synergy* models (*p*=0.95).

To obtain a measure of the overall fit between neural responses and behavioral performance, we computed the $R^2$ coefficient between the fMRI data and each behavioral model across voxels, subjects, and acquisition modalities. The group averages were 0.41 ± 0.06 for the *kinematic synergies*, 0.37 ± 0.03 for the *individual digits*, and 0.37 ± 0.06 for the *muscle synergies*. Therefore, 40.8% of the BOLD signal was accounted for by the *kinematic synergies*, whereas the two other behavioral models explained a relatively smaller fraction of the total variance.

## Functional neuroanatomy of kinematic hand synergies

The group analysis was performed only on the encoding results obtained from kinematic synergies, as this was the most successful model and the only one that performed above chance level across participants. The single-subject encoding results maps – containing only the voxels recruited during the procedure – were merged, with a threshold of *p*>0.33 to retain consistently informative voxels, overlapping in at least four participants.

The group-level probability map, which displays the voxels recruited in at least four subjects, consisted of a well-recognizable network of hand-related regions, specifically bilateral precentral cortex, supplementary motor area (SMA), ventral premotor and supramarginal areas, left inferior parietal and postcentral cortex (*Figure 1*; coordinates in *Supplementary file 1C*).

## Behavioral and neurofunctional stability of kinematic synergies and synergy-topic mapping

Since postural synergies were obtained in each subject independently, a procedure to assess the stability of the principal components (PCs) across participants was performed (see Materials and methods section). For visualization purposes, we focused on the first three PCs, which could explain more than 80% of the variance across the entire hand kinematic dataset, and were also highly consistent across participants (*Video 1*).

Accordingly to the aforementioned results, the first three kinematic PCs were mapped onto a flattened mesh of the cortical surface. This map displayed the fitting of each synergy within the voxels that were recruited by the encoding procedure across participants. *Figure 2* shows that the group kinematic synergies are represented in the precentral and postcentral cortex in distinct clusters that are arranged in a topographical *continuum* with smooth supero-inferior transitions. The procedure developed to assess the topographical arrangement of synergies (see Materials and methods) was statistically significant (C=0.19; *p*=0.038), indicating that anatomically close voxels exhibited similar synergy coefficients (see *Figure 2—figure supplement 1*).

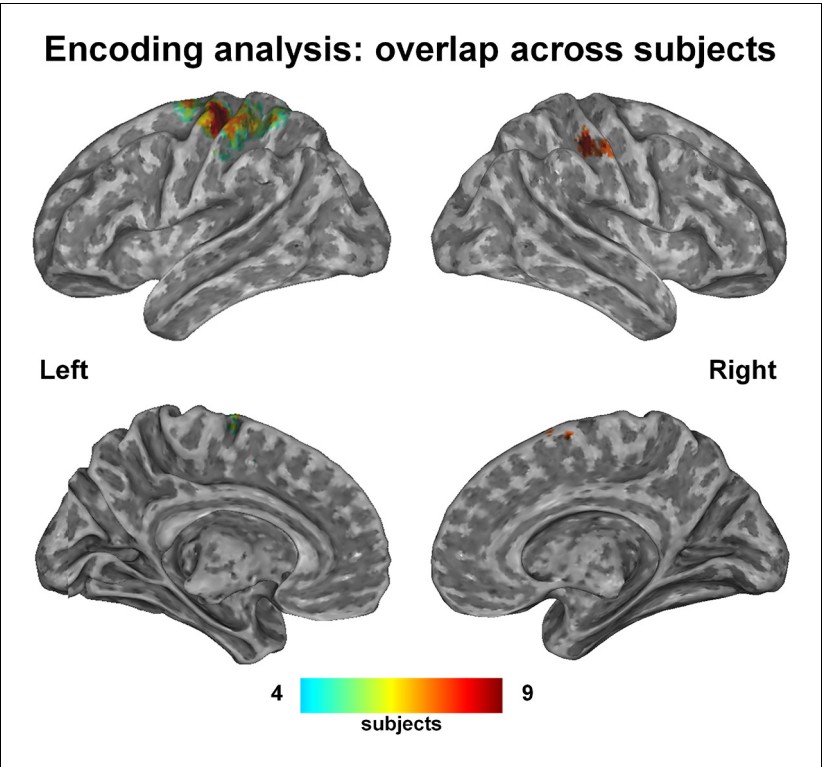

**Figure 1.** This probability map shows the voxels that were consistently engaged by the encoding procedure across subjects, i.e., those voxels whose activity was predictable on the basis of the kinematic synergies. A hand-posture- related network comprising the left primary and supplementary motor areas, the superior parietal lobe and the anterior part of intraparietal sulcus (bilaterally) was recruited with high overlap across subjects. Despite additional regions (i.e., Brodmann Area 6) resulted from the encoding analyses, they are not evident in the map due to their deep location.

The following source data is available for figure 1:

**Source data 1.** This compressed NIfTI file in MNI152 space represents the voxels that were recruited by the encoding procedure in more than three subjects.

**Source data 2.** his compressed NIfTI file in MNI152 space represents the Region of Interest chosen for encoding brain activity from the visual region, defined on the basis of a t-test of the overall brain activity (i.e., task versus rest condition) five seconds after the visual stimulus onset, corrected for multiple comparisons with False Discovery Rate (q<0.01).

## Representational similarity analysis (RSA) and multidimensional scaling (MDS)

Representational Spaces, drawn separately for the three models and fMRI data (using the activity from a region consistently activated across all the grasping movements), were compared at a single subject and group level to assess the similarity between each behavioral model and the neural content represented at a cortical level. All group correlations, both between fMRI and behavioral data and between behavioral models were highly significant ($p<0.0001$) (for details see *Supplementary file 1D,E* and *Figure 3—figure supplement 1*). Moreover, a MDS procedure was performed to represent data from kinematic synergies and fMRI BOLD activity. *Figure 3* shows the high similarity between these two spaces.

# Kinematic PC 1

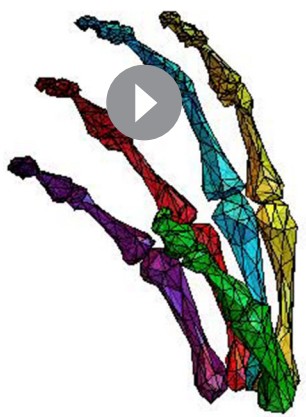

**Video 1.** This video shows the meaning of the kinematic synergies measured in this study, by presenting three movements from the minimum to the maximum values of kinematic synergies 1, 2, and 3, respectively, expressed as sets of twenty-four joint angles averaged across subjects. It can be observed that the first synergy modulates abduction-adduction and flexion-extension of both the proximal and distal finger joints, while the second synergy reflects thumb opposition and flexion-extension of the distal joints only. Maximizing the first synergy leads, therefore, to a posture resembling a power grasp, while the second one is linked to pinch movements directed towards smaller objects, and the third one represents movements of flexion and thumb opposition (like in grasping a dish or a platter) (*Santello et al., 1998*; *Gentner and Classen, 2006*; *Ingram et al., 2008*; *Thakur et al., 2008*).

## From kinematic PCs to brain activity, and back: hand posture reconstruction from brain activity

To confirm the presence of a neural representation of hand synergies at a cortical level and that that this information can be used to specifically control hand postures based on brain activity, we applied decoding methods as complementary approaches to encoding analyses (*Naselaris et al., 2011*). Hand posture (expressed as a matrix of 24 joints angles by 20 hand postures) was therefore predicted with a multiple linear regression procedure from fMRI data. Specifically, this procedure could reliably reconstruct the different hand postures across participants. The goodness-of-fit ($R^2$) between the original and reconstructed joint angle patterns related to single movements, averaged across subjects, ranged between 0.51 and 0.90 (*Supplementary file 1F*). Three hand plots displaying original and reconstructed postures from a representative subject are shown in *Figure 4*. Notably, this decoding attempt reveals that brain activity elicited by our task can effectively be used to reconstruct the postural configuration of the hand. Moreover, the rank accuracy procedure specifically designed to test the extent to which each decoded posture could be discriminated from the original ones yielded significant results in six of nine participants (*Supplementary file 1G*).

## The possible role of visual object presentation: control analyses

Since motor and premotor regions supposedly contains neuronal populations that respond to visual stimuli (*Kwan et al., 1985*; *Castiello, 2005*; *Klaes et al., 2015*), one may argue that the visual presentation of objects in the current experiment contributes to the synergy-based encoding of BOLD activity in those regions. To exclude this possibility, an encoding procedure using the *kinematic synergy* model was performed within the region of interest (ROI) chosen for RSA and posture reconstruction, using exclusively the neural activity related to visual object presentation, measured five seconds after the stimulus onset. The procedure was unsuccessful in all participants, thus indicating that the *kinematic synergy* information in motor and premotor regions was purely related to motor activity (*Supplementary file 1H*).

The encoding maps of kinematic synergies never included visual areas. Nonetheless, visual areas are likely to participate in the early stages of action preparation (*Gutteling et al., 2015*) and the motor imagery might have played a role during the task in the fMRI session. For this reason, we first defined a ROI by contrasting visual related activity after stimulus presentation and rest ($q<0.01$, FDR corrected), thus to isolate regions of striate and extrastriate cortex within the occipital lobe. Subsequently, an encoding analysis was performed similarly to the above-mentioned procedures. The results were at the chance level in seven out of nine participants (see *Supplementary file 1I*), suggesting that visual imagery processes in the occipital cortex did not retain *kinematic synergy* information.

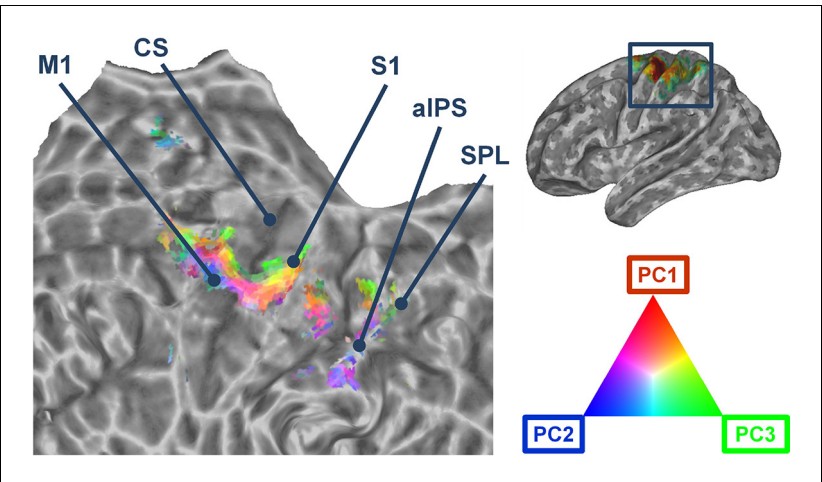

**Figure 2.** Cortical flattened map depicting the topographical organization of the first three synergies across primary motor, somatosensory, and parietal regions. The portion of cerebral cortex represented in the map corresponds to the area enclosed in the rectangle in the brain mesh (*top, right*). M1: Primary Motor Cortex. CS: Central Sulcus. S1: Primary Somatosensory cortex (postcentral gyrus). aIPS: anterior intraparietal sulcus. SPL: Superior Parietal lobule.

The following figure supplement is available for figure 2:

**Figure supplement 1.** Topography assessment: map and feature spaces.

## Discussion

Scientists have debated for a long time how the human hand can attain the variety of postural configurations required to perform all the complex tasks that we encounter in activities of daily living. The concept of *synergy* has been proposed to denote functional modules that may simplify the control of hand postures by simultaneously recruiting sets of muscles and joints. In the present study, by combining kinematic, EMG, and brain activity measures using fMRI, we provide the first demonstration that hand postural information encoded through kinematic synergies is represented within the

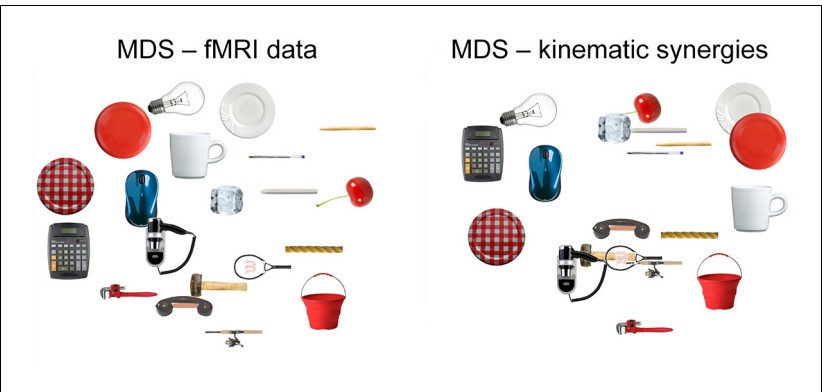

**Figure 3.** This picture displays the mMultidimensional sScaling (MDS) results for kinematic synergies (left) and fMRI brain activity (right). With the exception of few postures (e.g., dinner plate, frisbee and espresso cup) that were misplaced in the fMRI data with respect to the kinematic synergies representation, the other object-related postures almost preserved their relative distances.

The following figure supplement is available for figure 3:

**Figure supplement 1.** Average correlations between behavioral models and fMRI data.

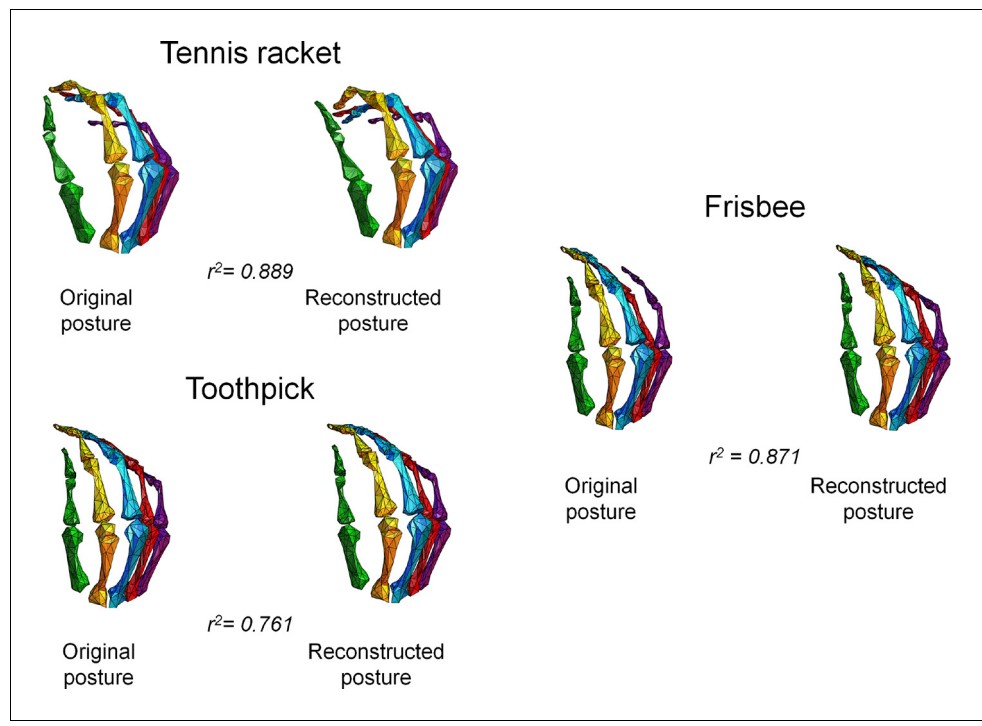

**Figure 4.** This picture represents the postures obtained from the fMRI data and those originally recorded through optical tracking. The figure shows three pairs of hand plots corresponding to three postures from a representative subject, and the goodness-of-fit between the original and decoded sets of joint angles. In these plots, the two wrist angles are not rendered.

The following source data and figure supplements are available for figure 4:

**Source data 1.** This compressed NIfTI file in MNI152 space represents the Region of Interest chosen for RSA and posture decoding, defined on the basis of a t-test of the overall brain activity (i.e., task versus rest condition), corrected for multiple comparisons with False Discovery Rate (q<0.05).

**Figure supplement 1.** Marker placement for kinematic hand posture data acquisition: The picture depicts the hand of a subject with the complete set of optical markers used to define hand posture through optical tracking.

**Figure supplement 2.** ROI used for performing RSA and posture decoding: This map represents the Region of Interest which contained all the voxels used for performing Representational Similarity Analysis and hand posture decoding.

---

cortical network controlling hand movements. Importantly, we demonstrate that kinematic synergies strongly correlate with the neural responses in primary and supplementary motor areas, as well as movement-related parietal and premotor regions. Furthermore, we show that kinematic synergies are topographically arranged in the precentral and postcentral cortex and represent meaningful primitives of grasping. Finally, the neural responses in sensorimotor cortex allow for a highly successful decoding of complex hand postures. Therefore, we conclude that the human motor cortical areas are likely to represent hand posture by combining few elementary modules.

## Kinematic synergies optimally predict behavioral outcomes and neurofunctional representations of distinct grasping-to-use motor acts

Validation of behavioral data was performed as the first stage of analysis to assess the information content and the discriminability of the postures from the kinematic or EMG data. This procedure showed that each posture could be successfully classified above chance level by *kinematic synergy,* *individual digit,* and *muscle synergy* models.

These results are highly consistent with the existent literature on synergies suggesting that just five PCs are sufficient to classify and reconstruct hand postures when computed only on hand kinematic data (*Santello et al., 1998*, *Santello et al., 2002*; *Gentner and Classen, 2006*), or both kinematic and EMG data (*Weiss and Flanders, 2004*; *Klein Breteler et al., 2007*). In the current work, we also demonstrate that kinematic synergies result in a higher discrimination accuracy of hand postures than individual digits and muscle synergies.

In addition, the encoding procedures on fMRI-based neural responses show that *kinematic synergies* are the best predictor of brain activity, with a significantly higher discrimination accuracy across participants, indicating that *kinematic synergies* are represented at a cortical level. Even if previous studies suggest that the brain might encode grasp movements as combinations of synergies in the monkey (*Overduin et al., 2012*), or indirectly in humans (*Gentner and Classen, 2006*; *Gentner et al., 2010*), to the best of our knowledge, no direct evidence has been presented to date for a functional validation and characterization of neural correlates of synergy-based models.

The results from RSA suggest that the three models used to predict brain activity may have similar, correlated spaces. However, each model provides a unique combination of weights for each posture across different dimensions (e.g., synergies or digits), thus resulting in distinct descriptions of the same hand postures. It should be noted that both the *individual digit* model and the *muscle synergy* model failed to predict brain activity in four and two participants, respectively. Thus, while they discriminated hand postures at a behavioral level, these models are clearly less efficient than the *kinematic synergy model* in predicting neural activity.

Finally, the descriptive procedures (RSA and MDS) were performed to assess the differences between the fMRI representational space and the single-model spaces. The results indicate a high similarity between fMRI and kinematic synergies, as reflected in the largely overlapping representations obtained from kinematic data and fMRI as depicted in *Figure 3*.

A recent study employed descriptive procedures (i.e., RSA) to demonstrate that similar movement patterns of individual fingers are reflected in highly correlated patterns of brain responses, that, in turn, are more correlated with kinematic joint velocities than to muscle activity, as recorded through high-density EMG (*Ejaz et al., 2015*). Our paper introduces a methodological and conceptual advancement. While, in Ejaz et al., full matrices of postural, functional or muscle data have been considered in the RSA, here we focused on descriptions with lower dimensionality, which lose only minor portions of information. Consequently, by showing that brain activity in motor regions can be expressed as a function of a few meaningful motor primitives that group together multiple joints, rather than as combinations of individual digit positions, our results suggest that a modular organization represents the basis of hand posture control.

## The functional neuroanatomy of kinematic synergies is embedded in motor cortical areas

The group probability maps of our study indicate that the regions consistently modulated by kinematic synergies, that include bilateral precentral, SMA and supramarginal area, ventral premotor, left inferior parietal and postcentral cortex, overlapped with a network strongly associated with the control of hand posture (*Castiello, 2005*).

Specifically, we show that the combination of five synergies, expressed as PCs of hand joint angles, predicts neural activity of M1 and SMA, key areas for motor control. While previous studies in humans showed differential activations in M1 and SMA for power and precision grip tasks (*Ehrsson et al., 2000*) and for different complex movements (*Bleichner et al., 2014*), to date no brain imaging studies directly associated these regions with synergy-based hand control.

Beyond primary motor areas, regions within parietal cortex are involved in the control of motor acts (*Grafton et al., 1996*). Inferior parietal and postcentral areas are engaged in higher-level processing during object interaction (*Culham et al., 2003*). Since grasping, as opposed to reaching movements, requires integration of motor information with inputs related to the target object, these regions may integrate the sensorimotor features needed to preshape the hand correctly (*Grefkes et al., 2002*; *Culham et al., 2003*). Consistently, different tool-directed movements were decoded from brain activity in the intraparietal sulcus (*Gallivan et al., 2013*), and it has been reported that this region is sensitive to differences between precision and power grasps (*Ehrsson et al., 2000*; *Gallivan et al., 2011*). The current motor task, even if performed with the dominant right hand only, also recruited motor regions of the right hemisphere. Specifically, bilateral

activations of SMA were often described during motor tasks (*Ehrsson et al., 2001*; *Ehrsson et al., 2002*) and a recent meta-analysis indicated a consistent recruitment of SMA in grasp type comparisons (*King et al., 2014*). Equally, a bilateral, but left dominant, involvement of intraparietal cortex for grasping has been reported (*Culham et al., 2003*).

Moreover, some authors have hypothesized recently that action recognition and mirror mechanisms may rely on the extraction of reduced representations of gestures, rather than on the observation of individual motor acts (*D'Ausilio et al., 2015*). The specific modulation of neural activity by kinematic synergies within the action recognition network seems in agreement with this proposition.

The map of voxels whose activity is modulated by postural synergies extends beyond the central sulcus to primary somatosensory cortex, suggesting a potential two-fold (sensory and motor) nature of hand synergies. Indeed, at least some subdomains (areas 2 and 3a) contain neurons that respond to multiple digits (*Iwamura et al., 1980*), despite the evidence supporting specific single finger representations in S1 (*Kaas, 1983*).

Finally, the width of our probability maps, measured on the cortical mesh, was ca. 1cm, which corresponds to the hand area, as defined by techniques with better spatial resolution, including ultra-high field fMRI or electrocorticography (ECoG) (*Siero et al., 2014*).

## Beyond the precision vs. power grasp dichotomy: synergy-based posture discrimination across participants

To exclude that the results from the encoding analysis can be driven by differences between classes of acts, i.e., precision or power grasps, rather than reflect the modulation of brain activity by kinematic synergies, the similarity between the 20 hand postures was evaluated in a pairwise manner. Specifically, the accuracy of the encoding model was estimated for each pair of distinct movements, unveiling the extent to which individual hand postures could be discriminated from each other based on their associated fMRI activity. In the result heat map (*Figure 5*), two clusters can be identified: one composed mainly by precision grasps directed towards small objects, and a second one composed mainly by power grasps towards heavy tools. The remaining postures did not cluster, forming instead a non-homogeneous group of grasps towards objects that could be either small (e.g., espresso cup) or large (e.g., jar lid, PC mouse).

These results indicate that goal-directed hand movements are represented in the brain in a way that goes beyond the standard distinction between precision and power grasps (*Napier, 1956*; *Ehrsson et al., 2000*). Other authors have proposed a possible 'grasp taxonomy' in which multiple, different types of grasps are described according to hierarchical criteria rooted on three main classes: precision, power and intermediate (*Feix et al., 2009*). By combining these three elementary grasps, it is possible to generate a wide number of postures. Notwithstanding the advancements of these taxonomies in describing hand posture, much less effort has been made to understand how the wide variety of human hand postures can be represented in the brain. Our results indicate that a synergy framework may predict brain activity patterns underlying the control of hand posture. Of note, the highest-ranked kinematic synergies can be clearly identified as grasping primitives: the first synergy modulates abduction-adduction and flexion-extension of both the proximal and distal finger joints, while a second synergy reflects thumb opposition and flexion-extension of the distal joints only. Maximizing the first synergy leads therefore to a posture resembling a power grasp, while the second one is linked to pinch movements directed towards smaller objects, and the third one represents movements of flexion and thumb opposition (like in grasping a dish or a platter) (*Santello et al., 1998*; *Gentner and Classen, 2006*; *Ingram et al., 2008*; *Thakur et al., 2008*) (*Video 1*). For this reason, the description of hand postures can benefit from reduction to combinations of few, meaningful synergies, which can provide more reliable results than clustering methods based on a small number of categories (*Santello et al., 2002*; *Ingram et al., 2008*; *Thakur et al., 2008*; *Tessitore et al., 2013*).

## How many hand synergies do humans have?

In the present study, we examined five hand postural synergies. This number was selected based on previous behavioral studies that showed that three and five PCs can account for at least 80% and 90% of the variance, respectively (*Santello et al., 1998*, *Santello et al., 2002*; *Weiss and Flanders, 2004*; *Gentner and Classen, 2006*; *Gentner et al., 2010*; *Overduin et al., 2012*). Indeed, a model

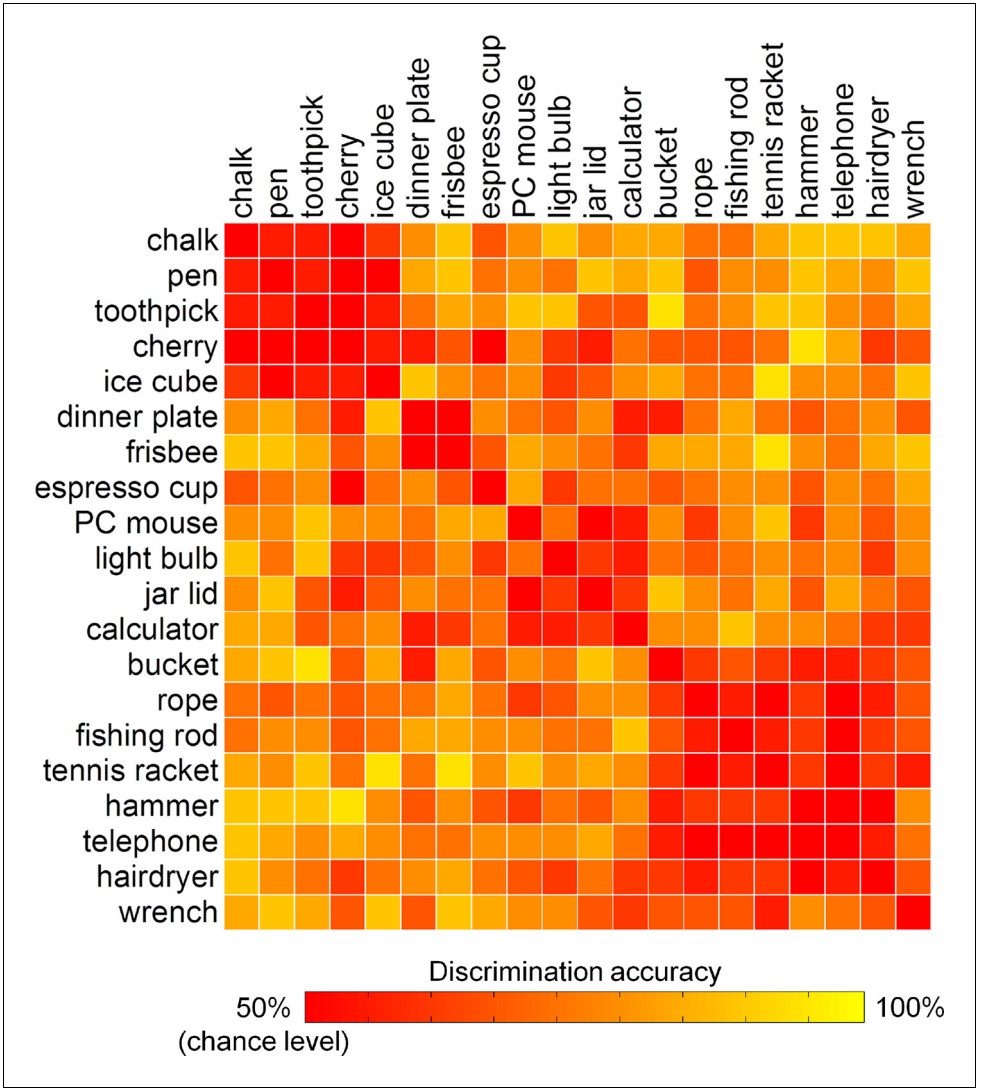

**Figure 5.** Discrimination accuracies for single postures as represented by kinematic synergies. Two clusters of similar postures are easily identifiable (i.e., precision grip and power grasps). However, other postures were recognized without showing an evident clustering, suggesting that the encoding procedure was not biased by a coarse discrimination of motor acts.

The following figure supplement is available for figure 5:

**Figure supplement 1.** Workflow of the encoding analysis.

with five synergies could successfully predict brain activation patterns. The first three synergies examined in the present study also show a high degree of stability as the order of the most relevant PCs is highly preserved across the nine participants. Moreover, the synergies described in the current study are consistent with those reported by other authors (*Santello et al., 1998*, *Santello et al., 2002*; *Gentner and Classen, 2006*; *Ingram et al., 2008*; *Thakur et al., 2008*), although a larger number of both postures and subjects would be required for the definitive characterization of the stability of hand postural synergies.

## A challenge to individual digit cortical representations? The functional topography of hand synergies

The first three synergies are displayed on a flattened map of the cortical surface in *Figure 2*. The map suggests that the PCs are topographically arranged, forming clusters with a preference for

each of the three synergies, separated by smooth transitions. This organization resembles that observed in the retinotopy of early visual areas (*Sereno et al., 1995*) or in auditory cortex as studied with tonotopic mapping (*Formisano et al., 2003*). This observation strongly suggests that primary motor and somatosensory brain regions may show specific, organized representations of synergies across the cortical surface. Such an observation is unprecedented, since the large number of previous studies adopted techniques, such as single cell recording (*Riehle and Requin, 1989*; *Zhang et al., 1997*) or intracortical microstimulation (ICMS) (*Overduin et al., 2012*), which can observe the activity of single neurons but do not capture the functional organization of motor cortex as a whole. Motor cortex has historically been hypothesized to be somatotopically organized in a set of sub-regions that control different segments of the body (*Penfield and Boldrey, 1937*). However, whereas subsequent work confirmed this organization (*Penfield and Welch, 1951*), a major critical point remains the internal organization of the single subregions (e.g., hand, leg or face areas). To date, a somatotopy of fingers within the hand area appears unlikely: as each digit is controlled by multiple muscles, individual digits may be mapped in a distributed rather than discrete fashion (*Penfield and Boldrey, 1937*; *Schieber, 2001*; *Graziano et al., 2002*; *Aflalo and Graziano, 2006*). An alternative view posits that movements are represented in M1 as clusters of neurons coding for different action types or goals (*Graziano, 2016*). In fact, mouse motor cortex is organized in clusters that encode different motor acts (*Brown and Teskey, 2014*). Similarly, stimulation of motor cortex in monkeys produces movements directed to stable spatial end-points (*Graziano et al., 2002*; *Aflalo and Graziano, 2006*) and may have a synergistic organization (*Overduin et al., 2012*). Recently, it has been demonstrated in both monkeys and humans that complex movements can be recorded from parietal as well as premotor and motor areas (*Aflalo et al., 2015*; *Klaes et al., 2015*; *Schaffelhofer et al., 2015*). Interestingly, a successful decoding can be achieved in those regions both during motor planning and execution (*Schaffelhofer et al., 2015*). These observations about the internal organization of motor cortex were demonstrated also in humans, revealing that individual representations of digits within M1 show a high degree of overlap (*Indovina and Sanes, 2001*) and that, despite digits may be arranged in a coarse ventro-dorsal order in somatosensory cortex, their representations are intermingled so that the existence of digit specific voxels is unlikely (*Ejaz et al., 2015*). In contrast, individual cortical voxels may contain enough information to encode specific gestures (*Bleichner et al., 2014*).

## Measuring synergies: back from brain signal to motor actions

Finally, we questioned whether the information encoded in M1 could be used to reconstruct hand postures. To this aim, each individual posture was expressed as a set of synergies that were derived from the fMRI activity on an independent cortical map. The results were reported as correlation values between the sets of joint angles originally tracked during kinematic recording and the joint angles derived from the reconstruction procedure. Overall, hand postures can be reconstructed with high accuracy based on the neural activity patterns. This result yields potential applications for the development of novel brain computer interfaces: for instance, previous studies demonstrated that neural spikes in primary motor cortex can be used to control robotic limbs used for performing simple or complex movements (*Schwartz et al., 2006*; *Schwartz, 2007*; *Velliste et al., 2008*). Previous studies in monkeys suggest that neural activity patterns associated with grasp trajectories can be predicted from single neuron activity in M1 (*Saleh et al., 2010*; *Saleh et al., 2012*; *Schaffelhofer et al., 2015*) and recently neuronal spikes have been associated with principal components (*Mollazadeh et al., 2014*). In humans, cortical activity obtained through intracranial recordings can be used to decode postural information (*Pistohl et al., 2012*) and proper techniques can even lead to decode EMG activity from fMRI patterns (*Ganesh et al., 2008*) or from ECoG signals (*Flint et al., 2014*). So far, decoding of actual posture from fMRI activity in M1 was possible at individual voxel level, albeit with simplified paradigms and supervised classifiers that identified only four different movements (*Bleichner et al., 2014*). In contrast, by proving that posture-specific sets of joint angles – expressed by synergy loadings – can be decoded from the fMRI activity, we show that information about hand synergies is present in functional data and can be even used to identify complex gestures. Other authors similarly demonstrated that a set of few synergies can describe hand posture in a reliable way, obtaining hand postures that correlated highly with those recorded with optical tracking (*Thakur et al., 2008*).

## Limitations and methodological considerations

While nine subjects may appear to be a relatively limited sample for a fMRI study, our study sample is comparable to that of most reports on motor control and posture (e.g., *Santello et al., 1998*; *Weiss and Flanders, 2004*; *Ingram et al., 2008*; *Thakur et al., 2008*; *Tessitore et al., 2013*; *Ejaz et al., 2015*) as well as to the sample size of fMRI studies that use encoding techniques, rather than univariate analyses (*Mitchell et al., 2008*; *Huth et al., 2012*). In addition, the data of our multiple experimental procedures (i.e., kinematic tracking, EMG, and fMRI) were acquired within the same individuals, so to minimize the impact of inter-subject variability and to facilitate the comparison between different models of hand posture. Finally, robust descriptive and cross-validation methods complemented single-subject multivariate approaches, which are less hampered by the number of participants than univariate fMRI procedures at group level.

A further potential criticism may involve the use of imagined objects – instead of real objects – as targets for grasping movements. The use of imagined objects allows to avoid confounding variables including grasping forces, difficulty in handling objects within a restricting environments, that could play a role in modulating motor acts. In previous behavioral reports, synergies were evaluated using contact with real objects (*Santello et al., 2002*) and participants could also explore them in an unconstrained manner instead of concentrating on single actions (e.g., grasping) (*Thakur et al., 2008*). Another study tracked hand motion across many gestures performed in an everyday life setting (*Ingram et al., 2008*). Interestingly, the dimensionality reduction methods were adopted with high consistency in these reports, despite the wide variety of experimental settings, and the first few PCs could explain most of the variance across a very wide number of motor acts. Moreover, when motor acts were performed toward both real and imagined objects, the results obtained from synergy evaluation were highly similar (*Santello et al., 2002*).

It can be argued that the better performance for *kinematic synergies* as compared to the other two alternative models may be due to the differences in the intrinsic signal and noise levels of the optical motion tracking and EMG acquisition techniques. Moreover, the *muscle synergy* model is inevitably simplified, since only a fraction of the intrinsic and extrinsic muscles of the hand can be recorded with surface EMG. Since all these factors may impact our ability to predict brain activity, we tested whether and to what extent different processing methods and EMG channel configurations could affect the performance of the *muscle synergy* model in discriminating single gestures and encoding brain activity. Therefore, we performed an additional analysis on an independent group of subjects, testing different processing methods and EMG channel configurations (up to 16 channels). The results, reported in the Appendix, demonstrate that EMG recordings with a higher dimensionality (*Gazzoni et al., 2014*; *Muceli et al., 2014*) or a different signal processing (*Ejaz et al., 2015*) do not lead to better discrimination results. These findings are consistent with previous reports (*Muceli et al., 2014*), and indicate that, in the current study, the worst performance of the muscle model relates more to the signal-to-noise ratio of the EMG technique per se, rather than to shortcomings of either the acquisition device or the signal processing methods adopted here.

While our data suggest that synergies may be arranged topographically on the cortical surface, the assessment of such a mapping is currently limited to the first three unrotated PCs. Additional studies are needed to investigate how topographical organization may be affected by the rotation of the principal components. Indeed, such an assessment requires the definition of stable population-level synergies to allow for the identification of optimally rotated components and to test their topographical arrangements across subjects; for this reason, it falls beyond the aims of the current study. Our work demonstrates that the topography of synergies, as defined as a spatial map of the first three PCs, is resistant to different arrangements; however, alternative configurations (rotated solutions within the PCA) can be encoded as well in sensorimotor cortical areas. The relatively low C index obtained in the mapping procedure and the total variance explained by the *kinematic synergy* model during the encoding procedure leave the door open to better models and different topographical arrangements.

## Beyond synergies: which pieces of information are also coded in the brain?

In summary, our results provide strong support for the representation of hand motor acts through postural synergies. However, this does not imply that synergies are the only way the brain encodes hand movements in primary motor cortex. In our data, only a portion (40%) of the total brain activity could be accounted for by kinematic synergies. Hand motor control results from complex interactions involving the integration of sensory feedback with the selection of motor commands to group of hand muscles. Similarly, motor planning is also a complex process, which requires selecting the desired final posture based on the contact forces required to grasp or manipulate an object. These elements must be continuously monitored to allow for on-line adaptation and corrections (*Castiello, 2005*). Previous studies demonstrated that only a small fraction of variance in M1 is related to arm posture (*Aflalo and Graziano, 2006*) and that grasping force can be efficiently decoded from electrical activity, suggesting that at least a subset of M1 neurons processes force-related information (*Flint et al., 2014*). In addition, motor areas can combine individual digit patterns on the basis of alternative non-synergistic or nonlinear combinations and the correlated activity patterns for adjacent fingers may depend on alternative mechanisms such as finger enslaving (*Ejaz et al., 2015*). It is likely that sensorimotor areas encode also different combinations of synergies, based – for instance – on the rotated versions of kinematic PCs: the encoding of synergies and of their rotated counterparts may represent a wider repertoire of motor primitives which can improve the flexibility and adaptability of modular control. Moreover, the information encoded may be related to the grasping action as a whole, not only to its final posture. Dimensionality reduction criteria can also be applied to hand posture over time, leading to time-varying synergies that encode complete preshaping gestures without being limited to their final position (*Tessitore et al., 2013*). This is consistent with EMG studies, which actually track muscle activity over the entire grasping trajectory (*Weiss and Flanders, 2004*; *Cheung et al., 2009*) and can add information about the adjustments performed during a motor act. Information about the temporal sequence of posture and movements may therefore be encoded in M1 and a different experimental setup is needed to test this hypothesis.

It should also be noted that studies in animal models bear strong evidence for a distributed coding of hand synergies beyond motor cortex, i.e., spinal cord (*Overduin et al., 2012*; *Santello et al., 2013*). The question about the role of M1 – i.e., whether it actually contains synergic information or simply act as a mere selector of motor primitives that are encoded elsewhere – still remains open. Our study provides a relatively coarse description of the role of M1 neurons. According to the redundancy principle, only a part of M1 neurons may be directly implied in movement or posture control (*Latash et al., 2007*), whereas the remaining neurons may deal with force production or posture adjustments and control over time, allowing for the high flexibility and adaptability which are peculiar features of human hand movements.

Altogether, the coding of motor acts through postural synergies may shed new light on the representation of hand motor acts in the brain and pave the way for further studies of neural correlates of hand synergies. The possibility to use synergies to reconstruct hand posture from functional activity may lead to important outcomes and advancements in prosthetics and brain-machine interfaces. These applications could eventually use synergy-based information from motor cortical areas to perform movements in a smooth and natural way, using the same dimensionality reduction strategies that the brain applies during motor execution.

## Materials and methods

### Subjects

Nine healthy volunteers (5F, age 25 ± 3 yrs) participated in the study. The subjects were right-handed according to the Edinburgh Handedness Inventory (*Oldfield, 1971*). All participants had normal or corrected-to-normal visual acuity and received a medical examination, including a brain structural MRI scan, to exclude any disorder that could affect brain structure or function.

### Experimental setup

The kinematic, EMG, and fMRI data were acquired during three separate sessions that were performed on different days, in a randomly alternated manner across participants. Eight of nine subjects

performed all the three sessions, while EMG data from one participant were not recorded due to hardware failure. Across the three sessions, participants were requested to perform the same task of grasp-to-use gestures towards 20 different virtual objects. A training phase was performed prior to the sessions to familiarize participants with the experimental task.

The kinematic and EMG experiments were performed to obtain accurate descriptions of the final hand posture. Three models of equal dimensions (i.e., five dimensions for each of the twenty postures) were derived from these two sessions: a *kinematic synergy model* based on PCA on kinematic data, an additional kinematic description which considers separately the displacements of each *individual digit* for each posture, and an EMG-based *muscle synergy model*. The models were first assessed using a machine-learning approach to measure their ability to discriminate among individual postures. The models were then used in a comparable method (i.e., encoding procedure) aimed at predicting the fMRI activity while subjects performed the same hand grasping gestures. Finally, fMRI activity was used to reconstruct the hand postures (i.e., decoding procedure).

## Kinematic experiment

The first experimental session consisted of kinematic recording of hand postures during the execution of motor acts with common objects. More specifically, we focused on the postural (static) component at the end of reach-to-grasp movements. Kinematic postural information was acquired with the model described in a previous study (*Gabiccini et al., 2013*), which is a fully parameterized model, reconstructed from a structural magnetic resonance imaging of the hand across a large number of postures (*Stillfried et al., 2014*). Such a model can be adapted to different subjects through a suitable calibration procedure. This model is amenable to in vivo joint recordings via optical tracking of markers attached to the skin and is endowed with a mechanism for compensating soft tissue artifacts caused by the skin and marker movements with respect to the bones (*Gustus et al., 2012*).

## Kinematic data acquisition

During the recordings, participants were comfortably seated with their right hand in a resting position (semipronated) and were instructed to lift and shape their right hand as to grasp a visually-presented object. Stimuli presentation was organized into trials in which pictures of the target objects were shown on a computer screen for three seconds and were followed by an inter-stimulus pause (two seconds), followed by an auditory cue that prompted the grasping movements. The interval between two consecutive trials lasted seven seconds. In each trial, subjects were requested to grasp objects as if they were going to use them, and to place their hands in the resting position once the movement was over. Twenty different objects, chosen from our previous report (*Santello et al., 1998*), were used in the current study (see *Supplementary file 1J* for a list).

The experiment was organized in five runs, each composed by twenty trials, in randomized order across participants. Therefore, all the grasp-to-use movements were performed five times. The experiment was preceded by a training session that was performed after the positioning of the markers. Hand posture was measured by an optical motion capture system (Phase Space, San Leandro, CA, USA), composed of ten stereocameras with a sampling frequency of 480 Hz. The cameras recorded the Cartesian positions of the markers and expressed them with reference to a global inertial frame and to a local frame of reference obtained by adding a bracelet equipped with optical markers and fastened to the participants' forearm. This allowed marker coordinates to be expressed with reference to this local frame. To derive the joint angles of the hand, other markers were placed on each bone (from metacarpal bones to distal phalanxes) and on a selected group of joints: thumb carpo-metacarpal (CMC), metacarpophalangeal (MCP) and interphalangeal (IP); index and middle MCPs; and all proximal interphalangeals (PIPs). This protocol is shown in *Figure 5—figure supplement 1* and a full list of markerized joints and their locations can be found in *Supplementary file 1K* and in *Gabiccini et al., 2013*.

The placement of the markers was performed according to the model described in *Gabiccini et al., 2013*, which consists of 26 Degrees of Freedom (DoFs), 24 pertaining to the hand and 2 to the wrist. The wrist markers were not used in subsequent analyses. The marker configuration resembles a kinematic tree, with a root node corresponding to the Cartesian reference frame, rigidly fastened to the forearm, and the leaves matching the frames fixed to the distal phalanxes (PDs) of the five digits, as depicted in the first report of the protocol (*Gabiccini et al., 2013*).

## Kinematic data preprocessing

First, the frame rate from the ten stereocameras was downsampled to 15 Hz. After a subject-specific calibration phase, which was performed to extract the geometric parameters of the model and the marker positions on the hand of each participant, movement reconstruction was performed by estimating all joint angles at each sample with an iterative extended Kalman filter (EKF) which takes into account both measurements explanation and closeness to the previous reconstructed pose (see *Gabiccini et al., 2013* for further details).

Once all trials were reconstructed, the posture representing the final grasping configuration was selected through direct inspection. The final outcome of this procedure was a 24 x 100 matrix for each subject, containing 24 joint angles for 20 objects repeated five times.

## Kinematic model

The kinematic data from each subject were analyzed independently. First, the hand postures were averaged across five repetitions for each object, after which the data matrix was centered by subtracting, from each of the 20 grasping movements, the mean posture calculated across all the motor acts. Two different models were obtained from the centered matrix. The first was a *kinematic synergy* model, obtained by reducing the dimensionality with a PCA on the 20 (postures) by 24 (joint angles) matrix and retaining only the first five principal components (PCs). In this way, the postures were projected onto the components space, hence obtaining linear combinations of synergies.

To obtain an alternative individual digit model, defined on a somatotopic basis, the displacement of *individual digits* was also measured (*Kirsch et al., 2014*). Briefly, the displacement of each finger for the twenty single postures was obtained by calculating the sum of the single joint angles within each digit and gesture, again excluding wrist DoFs.

The analyses of all the sessions were carried out using MATLAB (MathWorks, Natick, MA, USA), unless stated otherwise

## EMG experiment

The second session consisted of a surface electromyography acquisition (EMG) during the execution of grasp-to-use acts performed towards the same imagined objects presented during the kinematic experiment.

## EMG acquisition

EMG signals were acquired from five different muscles using self-adhesive surface electrodes. The muscles used for recording were: *flexor digitorum superficialis* (FDS), *extensor digitorum communis* (EDC), *first dorsal interosseus* (FDI), *abductor pollicis brevis* (APB), and *abductor digiti minimi* (ADM). The individuation of the sites for the recording of each muscle was performed according to the standard procedures for EMG electrode placement (*Hermens et al., 1999*; *Hermens et al., 2000*). The skin was cleaned with alcohol before the placement of electrodes.

Participants performed the same tasks and protocol used in the kinematic experiment, i.e., visual presentation of the target object (three seconds), followed by an inter-stimulus interval (two seconds), an auditory cue to prompt movement, and an inter-trial interval (seven seconds). The experiment was divided into runs that comprised the execution of grasping actions towards all the 20 objects, in randomized order. Participants performed six runs. Each gesture was therefore repeated six times.

EMG signals were recorded using two devices (Biopac MP35 for four muscles; Biopac MP150 for the fifth muscle) and Kendall ARBO 24-mm surface electrodes, placed on the above-mentioned muscles of the participants' right arm. EMG signals were sampled at 2 kHz.

## EMG model

First, EMG signals were resampled to 1 kHz and filtered with a bandpass (30–1000 Hz) and a notch (50 Hz) filter. For each channel, each trial (defined as a time window of 2500 samples) underwent the extraction of 22 primary time-domain features, chosen from those that are most commonly used in EMG-based gesture recognition studies (*Zecca et al., 2002*; *Mathiesen et al., 2010*; *Phinyomark et al., 2010*; *Tkach et al., 2010*; see *Chowdhury et al., 2013* for a review). Additional second-order features were obtained from the first features, computing their signal median, mean

absolute deviation (MAD), skewness, and kurtosis. A complete list of the EMG features we used can be found in *Supplementary file 1L*.

A muscle model was derived from the chosen features as follows: first, the pool of 410 features (82 for 5 channels) was reduced to its five principal components. The 1 x 5 vectors describing each individual movement were averaged across the six repetitions. This 20 (movements) x 5 (synergies) matrix represented the *muscle synergy* model for the subsequent analyses.

## Models validation

To verify that the three models (*kinematic synergies, individual digit*, and *muscle synergies*) were able to accurately describe hand posture, their capability to discriminate between individual gestures was tested. To this purpose, we developed a rank accuracy measure within a leave-one-out cross-validation procedure, as suggested by other authors to solve complex multiclass classification problems (*Mitchell et al., 2004*). For each iteration of the procedure, each repetition of each stimulus was left out (*probe*), whereas all other repetitions (*test set*) were averaged. Then, we computed PCA on the data from the *test set*. The PCA transformation parameters were applied to transform the *probe* data in a leave-one-repetition-out way. Subsequently, we computed the Euclidean distance between the *probe* element and each element from *test* dataset. These distances were sorted, generating an ordered list of the potential gestures from the most to the least similar. The rank of the *probe* element in this sorted list was transformed in a percentage accuracy score. The procedure was iterated for each target gesture and repetition of the same grasping movement. The accuracy values were first averaged across repetitions and then across gestures, resulting in one averaged value for each subject. In this procedure, if an element is not discriminated above chance, it may fall in the middle of the ordered list (around position #10), which corresponds to an accuracy of 50%. For this reason, the chance level is always 50%, regardless of the number of gestures under consideration, while 100% of accuracy indicated that the correct gesture in the sorted list retained the highest score (i.e., the lowest distance, first ranked) across repetitions and participants.

The accuracy values were then tested for significance against the null distribution of ranks obtained from a permutation test. After averaging the four repetitions within the *test* set, the labels of the elements were shuffled; then, the ranking procedure described above was applied. The procedure was repeated 10,000 times, generating a null distribution of accuracies; the single-subject accuracy value was compared against this null distribution (one-sided rank test). This procedure was applied to the three models extracted from kinematic and EMG data, obtaining a measure of noise and stability across repetitions and each posture, as described by the three different approaches. Such validation procedure was therefore a necessary step to measure the information content of these three models before testing their ability to predict the fMRI signal.

## Individuation of the optimal number of components

The extraction of postural or muscle synergies from kinematic and EMG data was based on a PCA applied to the matrices of sensor measures or signal features, respectively. For the analyses performed here, we chose models based on the first five principal components that were shown to explain more than 90% of the variance in previous reports, even if those models were applied on data with lower dimensionality (*Santello et al., 1998*; *Weiss and Flanders, 2004*; *Gentner and Classen, 2006*). Moreover, an additional model was obtained from the postural data, thus leading to three different models with the same dimensionality (five dimensions): a *kinematic synergy* model (based on PCA applied to joint angles), an *individual digit* model (based on the average displacement of the digits), and a *muscle synergy* model (based on PCA applied to EMG features). However, to verify that the procedures applied here to reduce data dimensionality yielded the same results of those applied in previous works, we performed PCA by retaining variable numbers of components, from 1 to 10, and applied the above-described ranking procedure to test the accuracy of all data matrices. The plots of the accuracy values as a function of the number of PCs can be found in *Supplementary files 1M* and in *Figure 6*. The result of this analysis confirmed that the present data are consistent with the previous literature. The same testing procedure was also applied to the *individual digit* model by computing the rank accuracies for the full model (five components) and for the reduced models with 1 to 4 PCs.

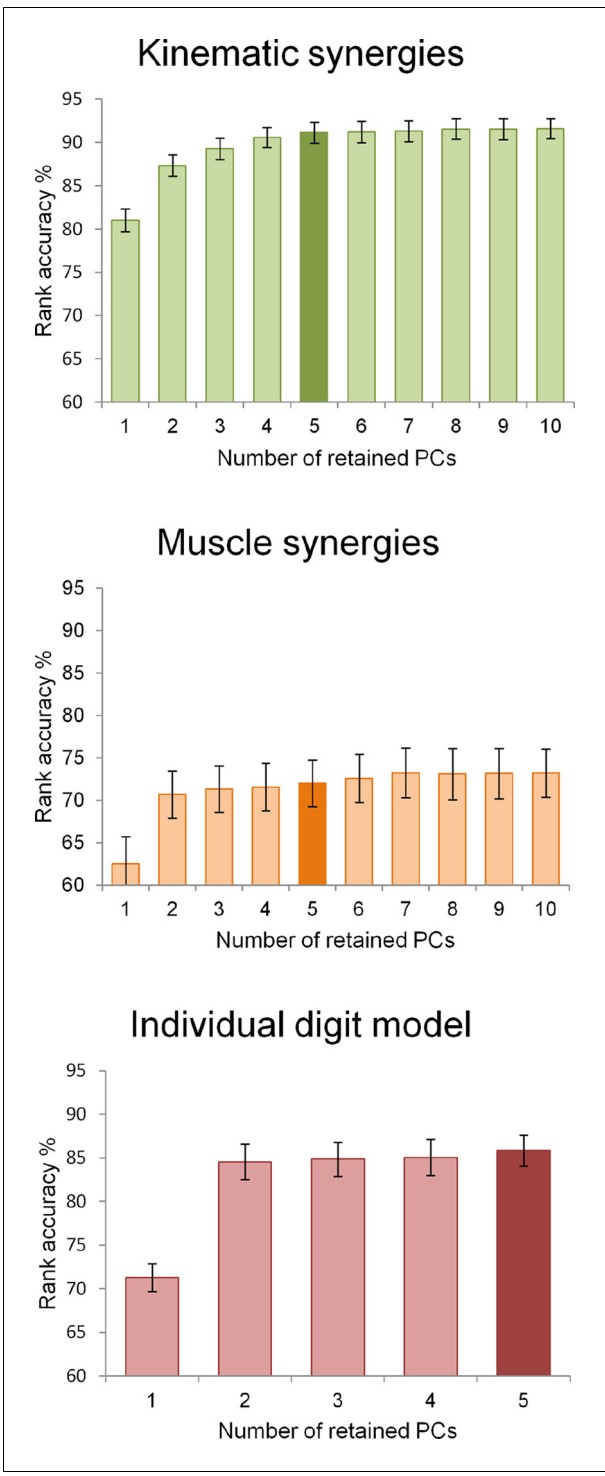

**Figure 6.** The three graphs display the rank accuracy values as a function of the dimensionality (i.e., the number of retained PCs) of each behavioral model. The two models derived from kinematic and EMG data (upper and middle graphs, respectively) have a number of synergies ranging from 1 to 10 while the individual digit model (lower) had 1 to 5 retained PCs. Darker bar colors indicate the dimensionality chosen for encoding brain functional data.

## fMRI experiment

In the third session, fMRI was used to record the brain activity during the execution of grasp-to-use acts with the objects presented during the previous experiments.

## fMRI acquisition

Functional data were acquired with a 3.0 Tesla GE Signa scanner (GE, Milwaukee, WI, USA), equipped with an 8-channel head-only coil. A Gradient-Echo echo-planar sequence was used, with an acquisition matrix of 128 x 128, FOV = 240 x 240 mm, Repetition Time (TR) = 2.5 s, Time of Echo (TE) = 40 ms, Flip Angle (FA) = 90°. Each volume comprised 43 3 mm-thick slices and the resulting voxel size was 1.875 x 1.875 x 3 mm. Additional anatomical images were also acquired with a high-resolution T1-weighted Fast Spoiled Gradient Recalled sequence (FSPGR) with 1 mm$^3$ isotropic voxels and a 256 x 256 x 0.170 mm$^3$ field-of-view; TR = 8.16 s, TE = 3.18 ms, FA = 12°. Head motion was minimized with foam pads.

The task design was identical to that used in previous sessions. Specifically, participants had to shape their hand as if grasping one of the twenty visually-presented objects. In the current session, the subjects were asked to perform only the hand preshaping, limiting the execution of reaching acts with their arm or shoulder, since those movements could easily cause head motion. The day before MRI, all subjects practiced movements in a training session.

The paradigm was composed of five runs, each consisting of 20 randomized trials. Each trial consisted of a visual presentation of the target object (2.5s), an inter-stimulus pause (5 s) followed by an auditory cue to prompt movements, and an inter-trial interval (12.5s). The functional runs had two periods of rest (15 s) at their beginning and end to measure baseline activity. The total duration was 6 min and 10 s (172 time points). The total scanning time was about 40 min.

In all sessions, visual stimuli were black and white pictures of the target objects, with a normalized width of 500 pixels. The auditory cue was an 800 Hz sound lasting 150 ms. The experimental paradigm was handled by the software package Presentation® (Neurobehavioral System, Berkeley, CA, http://www.neurobs.com) using a MR-compatible visual stimulation device (VisuaStim, Resonance Technologies, Northridge, CA, USA; dual display system, 5″, 30° of horizontal visual field, 800x600 pixels, 60 Hz) and a set of MR-compatible headphones for stimuli delivery.

## fMRI preprocessing

The initial steps of fMRI data analysis were performed with the AFNI software package (*Cox, 1996*). All volumes within each run were temporally aligned (3dTshift), corrected for head motion by registering to the fifth volume of the run that was closer in time to the anatomical image (3dvolreg) and underwent a spike removal procedure to correct for scanner-associated noise (3dDespike). A spatial smoothing with a Gaussian kernel (3dmerge, 4 mm, Full Width at Half Maximum) and a percentage normalization of each time point in the run (dividing the intensity of each voxel for its mean over the time series) were subsequently performed. Normalized runs were then concatenated and a multiple regression analysis was performed (3dDeconvolve). Each trial was modeled by nine tent functions that covered its entire duration from its onset up to 20 s (beginning of the subsequent trial) with an interval of 2.5 s. The responses associated with each movement were modeled with separate regressors and the five repetitions of the same trial were averaged. Movement parameters and polynomial signal trends were included in the analysis as regressors of no interest. The *t*-score response images at 2.5, 5, and 7.5 s after the auditory cue were averaged and used as estimate of the BOLD responses to each grasping movement compared to rest.

The choice to average three different time points for the evaluation of BOLD response was justified by the fact that such a procedure leads to simpler encoding models for subsequent analyses and that the usage of tent functions is a more explorative procedure that is not linked to an exact time point. For this reason, we could obtain an estimation of brain activity that is more linked to the motor act than to the visual presentation of the target object by concentrating only on a restricted, late time interval. This approach – or similar ones – has also been used by other fMRI studies (*Mitchell et al., 2008*; *Connolly et al., 2012*).

The coefficients, averaged related to the 20 stimuli of each subject, were transformed to the standard MNI 152 space. First FMRIB Nonlinear Image Registration Tool (FNIRT) was applied to the anatomical images to register them in the standard space with a resolution of 1 mm$^3$ (*Andersson et al.,*

*2007*). The matrix of nonlinear coefficients was then applied to the BOLD responses, which were also resampled to a resolution of 2 x 2 x 2 mm.

## fMRI single-subject encoding analysis

To identify the brain regions whose activity co-varied with the data obtained from the three models – kinematic, EMG synergies, and individual digits – a machine learning algorithm was developed, based on a modified version of the multiple linear regression encoding approach first proposed by Mitchell and colleagues (*Mitchell et al., 2008*). This procedure is aimed at predicting the activation pattern for a stimulus by computing a linear combination of synergy weights obtained from the behavioral models (i.e., Principal Components) with an algorithm previously trained on the activation images of a subset of stimuli (see *Figure 5—figure supplement 1*). The procedure consisted in 190 iterations of a leave-two-out cross-validation in which the stimuli were first partitioned in a training set (18 stimuli) and a test set with the two left-out examples. The sample for the analysis was then restricted to the 5000 voxels with the best average BOLD response across the 18 stimuli in the training set (expressed by the highest *t*-scores). For each iteration, the model was first trained with the vectorized patterns of fMRI coefficients of 18 stimuli associated with their known labels (i.e., the target objects). The training procedure employed a least-squares multiple linear regression to identify the set of parameters that, if applied to the five synergy weights, minimized the squared error in reconstructing the fMRI images from the training sample. After training the model, only the 1000 voxels that showed the highest $R^2$ (a measure of fitting between the matrix of synergy weights and the training data) were retained. A cluster size correction (nearest neighbor, size = 50 voxels) was also applied to prune small, isolated clusters of voxels. The performance of the trained model was then assessed in a subsequent decoding stage by providing it with the fMRI images related to the two unseen gestures and their synergy weights, and requiring it to associate an fMRI pattern with the label of one of the left-out stimuli. The procedure was performed within the previously chosen 1000 voxels and accuracy was assessed by considering the correlation distance between the predicted and real fMRI patterns for each of the two unseen stimuli. This pairwise procedure led therefore to a number of correctly predicted fMRI patterns ranging from 0 to 2 with a chance level of 50%. This cross-validation loop was repeated 190 times, leaving out all the possible pairs of stimuli. Therefore, the results consisted of an overall accuracy value – the percentage of fMRI patterns correctly attributed, which is an expression of the success of the model in predicting brain signals – and a map of the voxels that were used in the procedure – i.e., the voxels whose signal was predictable on the basis of the synergy coefficients. Every voxel had a score ranging from 0 (if the voxel was never used) to a possible maximum of 380 (if the voxel was among the 1000 with the highest $R^2$ and the two left-out patterns could be predicted in all the 190 iterations). The encoding analysis was performed in separate procedures for each model – i.e., *kinematic* and *muscle synergies* and *individual digit*. We obtained, therefore, three sets of accuracy values and three maps of the most used voxels for each subject. These results, which displayed the brain regions whose activity was specifically modulated by the grasping action that was performed inside the scanner, were subsequently used for building the group-level probability maps (see below).

## Assessment of the accuracy of the encoding analysis

The single-subject accuracy was tested for significance against the distribution of accuracies generated with a permutation test within the above-defined encoding procedure. Permutation tests are the most reliable and correct method to assess statistical significance in multivariate fMRI studies (*Schreiber and Krekelberg, 2013*; *Handjaras et al., 2015*). The null distribution of accuracies was built with a loop in which the model was first trained with five randomly chosen synergy weights that were obtained by picking a random value out of the 18 (one for each gesture) in each column of the matrix of synergies. The trained model was subsequently tested on the two left-out images. The procedure was repeated 1000 times, leading to a null distribution of 1000 accuracy values against which we compared the value obtained from the above-described encoding method. Similarly to the encoding analysis, we did not use either the fMRI images or the synergy weights of the two test stimuli for training the model. The left-out examples were therefore tested by an algorithm that had been trained on a completely independent data sample. The weights were shuffled only *within column*: this procedure yielded vectors of shuffled weights with the same variance as the actual

kinematic PCs, even though those vectors were no longer orthogonal. Permutation tests were performed separately for each subject with the three data matrices. Each single-subject accuracy was therefore tested against the null distribution of accuracy values obtained from the same subject data (one-sided rank test).

## Group-level probability maps

A group map displaying the voxels that were consistently recruited across subjects was obtained for the *kinematic synergy* model. The single-subject maps achieved from the encoding analysis, which display the voxels recruited by the encoding procedure in each subject, were first binarized by converting non-zero accuracy values to 1, then summed to obtain an across-subjects overlap image. Moreover, a probability threshold of these maps (*p*>0.33) was applied on the maps to retain voxels in which the encoding procedure was successful in at least four out of the nine subjects (*Figure 1*).

## Discrimination of single postures by fMRI data

The accuracies of pairwise discrimination of postures, achieved during the decoding stage of the encoding procedure, were combined across subjects, so to identify the postures that could be discriminated with the highest accuracy based on their associated BOLD activity. The results were displayed as a heat map (*Figure 5*), with a threshold corresponding to the chance level of 50%.

## Assessment of kinematic synergies across subjects

To evaluate whether the synergies computed on kinematic data from our sample would allow for a reliable reconstruction of hand posture, we needed to verify that these synergies are consistently ranked across individuals. Therefore, we used Metric Pairwise Constrained K-Means (*Bilenko et al., 2004*), a method for semi-unsupervised clustering that integrates distance function and constrained classes. We used the weights of the first three kinematic synergies for the 20 gestures in each subjects as input data and arranged the set of 27 20-items vectors into three classes with nine synergies that showed the higher similarity (see *Supplementary file 1N*). This analysis was limited to the first three PCs since previous reports (*Santello et al., 1998*; *Gentner and Classen, 2006*) suggest that they may constitute a group of "core synergies", with a cumulative explained variance greater than 80%. This analysis was performed only on the synergies obtained from the *kinematic synergy* model, which was able to outperform both the *individual digit* and *muscle synergy* models in terms of encoding accuracy percentages on fMRI data.

To facilitate the interpretation of the first kinematic PCs as elementary grasps, we plotted the time course of the corresponding hand movements. The plots are 2s-long videos showing three movements from the minimum to the maximum values of PCs 1, 2 and 3, respectively, expressed as sets of 24-joint angles averaged across subjects (*Video 1*).

## Cortical mapping of the three group synergies

The three group synergies were studied separately, computing the single correlations between each PC and the fMRI activation coefficient. This correlation estimated the similarity between the activity of every voxel for the 20 grasping acts and the weights of each single synergy. The coefficient of determination ($R^2$) for each synergy was averaged across participants to achieve a measurement of group-level goodness of fit. The overlap image between the group-level probability map and the goodness of fit for each synergy was then obtained and mapped onto a flattened mesh of the cortical surface (*Figure 2*). The AFNI SUMA program, the BrainVISA package and the ICBM MNI 152 brain template (*Fonov et al., 2009*) were used to render results on the cortical surface (*Figure 1* and *2*).

To provide a statistical assessment of the orderly mapping of synergies across the regions recruited by the encoding procedure, a comparison between the map space and the feature space was performed (*Goodhill and Sejnowski, 1997*; *Yarrow et al., 2014*). The correlation of the two spaces is expressed by an index (C parameter) that reflects the similarity between the arrangement of voxels in space and the arrangement of their information content: high values indicate that voxels which contain similar information are also spatially close, suggesting a topographical organization. The map space was derived measuring the standardized Euclidean distance between each voxel position in the grid. The feature space was computed using the standardized Euclidean distance

between the three synergy weights, as defined by their $R^2$, for each voxel and averaged across subjects according to the classes described in the sections *Assessment of kinematic synergies across subjects* and *Cortical mapping of the three group synergies*. The C parameter was achieved by computing the Pearson correlation between the map space and the feature space (*Yarrow et al., 2014*). An ad-hoc statistical test was developed to assess the existence of the topography. A permutation test was performed generating a null-distribution of C values by correlating the map space with feature spaces obtained by averaging the three synergies across subjects with different random combinations (10,000 iterations). The *p*-value was calculated by comparing the null-distribution with the C parameter obtained with the cortical mapping (one-sided rank test).

## Representational similarity analysis (RSA) and multidimensional scaling (MDS)

Representational content measures (*Kriegeskorte et al., 2008a*; *Kriegeskorte and Kievit, 2013*) were carried out to explore the information that is coded in the regions activated during the execution of finalized motor acts. Representational spaces (RSs) are matrices that display the distances between all the possible pairs of neurofunctional or behavioral measures, informing us about the internal similarities and differences that can be evidenced within a stimulus space. By computing a second-order correlation between single model RSs we can evaluate both the similarity between the information carried by the single behavioral models (kinematic, individual digits and EMG) and between behavioral data and brain activity as measured by fMRI.

RSA was therefore performed within a subset of voxels that were consistently activated by the task. A Region of Interest (ROI) was derived from the fMRI data by performing a *t*-test (AFNI program 3dttest++) that compared the mean brain activity at 2.5, 5, and 7.5 s after the auditory cue and the activity at rest. The results were corrected for False Discovery Rate (*Benjamini and Hochberg, 1995*; *p*<0.05) (*Figure 4—figure supplement 2*). Afterwards, the *t*-scores relative to each voxel within the ROI were normalized by subtracting the mean across-stimulus activation of all the voxels in the ROI and dividing the value by the standard deviation (*z*-score normalization). PCA was performed to reduce the BOLD activity of the voxels in the ROI to the first five principal components. Activation pattern RSs were then obtained for each subject by calculating the Euclidean distance between the PCs of all the possible pairs of stimuli (*Edelman et al., 1998*; *Kriegeskorte et al., 2008b*; *Haxby et al., 2014*). Model RSs were similarly computed for the three types of postural data. This procedure led to a set of brain activity RSs and three sets of model RSs for *kinematic synergy*, individual digit, and *muscle synergy* models, respectively. The single subject RSs were averaged to obtain a unique group RS for each model.

Since we were interested in identifying the similarities and differences between the information expressed by the behavioral models and the information encoded in the brain, we estimated Pearson correlation separately between the fMRI-based RS and each model RS (*Kriegeskorte et al., 2008a*, *2008b*; *Devereux et al., 2013*). Moreover, to study the possible specific relations between the behavioral models, additional pairwise correlations between the three model RSs were also performed.

These correlations were tested with the Mantel test by randomizing the twenty stimulus labels and computing the correlation. This step was repeated 10,000 times, yielding a null distribution of correlation coefficients. Subsequently, we derived the *p*-value as the percent rank of each correlation within this null distribution (*Kriegeskorte et al., 2008a*). The correlations were also estimated between single-subject RSs.

In addition, a MDS procedure, using standardized Euclidean distance, metric stress criterion and Procrustes alignment (*Kruskal and Wish, 1978*) was performed to represent the kinematic synergies and the patterns of BOLD activity across subjects (*Figure 3*).

## Decoding of hand posture from fMRI data

Additionally, the fMRI data were used to decode hand postures from stimulus-specific brain activity.

This procedure was performed using fMRI coefficients to obtain a set of 24 values, each representing the distances between adjacent hand joints, which could then be used to plot hand configuration. To this purpose, we first run a PCA on the fMRI data, using the voxels within the mask obtained for the RSA and MDS (see above and *Figure 4—figure supplement 2*) to avoid any

possible selection bias; with this procedure, the dimensionality of the data was reduced to the first five dimensions, as previously done for kinematic and EMG data.

Then, a multiple linear regression was performed within a leave-one-stimulus-out procedure by using the matrix of postural coefficients as predicted data and the reduced fMRI matrix as predictor. This allowed for the reconstruction of the coefficients of the left-out posture, yielding a matrix with 20 rows (postures) and 24 columns (joint angles). Finally, we estimated the goodness of fit ($R^2$) between the reconstructed data and the original postural matrices recorded with the optical tracking system, both subject-wise (i.e., computing the correlation of the whole matrices) and posture-wise (i.e., computing the correlation of each posture vector). In addition, the decoding performance was assessed using a rank accuracy procedure (similar to those performed in the behavioral analyses) in which each reconstructed posture was classified against those originally recorded during the kinematic experiment. The accuracy values were tested against the null distribution generated by a permutation test (10,000 iterations). The reconstructed data were then plotted, using custom code written in MATLAB and Mathematica 9.0 (Wolfram Research, Inc., Champaign, IL, USA) (*Figure 4*).

## Acknowledgements

We thank Mirco Cosottini and Luca Cecchetti for help with data collection, technical assistance, and critical discussions; Arash Ajoudani and Alessandro Altobelli for their help with additional experiments.

## Additional information

### Funding

| Funder | Grant reference number | Author |
| --- | --- | --- |
| European Research Council | ERC-291166 SoftHands | Matteo Bianchi<br>Hamal Marino<br>Marco Gabiccini<br>Andrea Guidi<br>Enzo Pasquale Scilingo<br>Antonio Bicchi |
| European Research Council | ERC-291166 The Hand Embodied | Andrea Leo<br>Giacomo Handjaras<br>Matteo Bianchi<br>Hamal Marino<br>Marco Gabiccini<br>Andrea Guidi<br>Enzo Pasquale Scilingo<br>Pietro Pietrini<br>Antonio Bicchi<br>Marco Santello<br>Emiliano Ricciardi |

The funders had no role in study design, data collection and interpretation, or the decision to submit the work for publication.

### Author contributions

AL, Conception and design, Acquisition of data, Analysis and interpretation of data, Drafting or revising the article; GH, MB, Acquisition of data, Analysis and interpretation of data, Drafting or revising the article; HM, AG, Acquisition of data, Analysis and interpretation of data; MG, EPS, Conception and design, Analysis and interpretation of data; PP, MS, ER, Conception and design, Analysis and interpretation of data, Drafting or revising the article; AB, Conception and design, Drafting or revising the article

### Ethics

Human subjects: This study was approved by the Ethical Committee at the University of Pisa, Italy. Participants received a detailed explanation of all the study procedures and risks and provided a written informed consent according to the protocol approved by the University of Pisa Ethical Committee (1616/2003). All participants retained the right to withdraw from the study at any moment.

# Additional files

## Supplementary files

• Supplementary file 1. (A) Single subject rank accuracy values. Values of rank accuracy, measured with the leave-one-stimulus-out procedure, for the nine subjects, with the *p*-value obtained from the permutation test (10000 iterations). The comparison between the performance values indicate that the *kinematic synergy* model was significantly better than both the *individual digit* and *muscle synergy* models (Wilcoxon signed-rank test, *p*=0.0078), and the *individual digit* model was significantly more informative than the *muscle synergy* model (*p*=0.0156).

(B) Single subject encoding accuracy values. The accuracy of predicting brain activity from the behavioral models (*kinematic synergy*, *individual digit* and *muscle synergy* models), obtained with the cross-validation procedure, is reported here for each subject, along with the chance levels derived from the permutation tests, the threshold at *p*=0.05 and the actual p-value obtained from the tests against the null distributions of accuracies. The accuracy values reported in red are not significant. The comparisons between individual accuracy values, performed using Wilcoxon signed-rank tests, show that the *kinematic synergy* model outperformed both the *individual digit (p*=0.0234) and the *muscle synergy (p*=0.0391) models, whereas no significant difference was found between the *individual digit* and *muscle synergy* models (*p*=0.9453).

(C) Size and coordinates of the clusters of greatest overlap between subjects. This table reports the regions that were consistently recruited across subjects (p>0.33, 4 out of 9 subjects). The region names are reported alongside with their size and with the coordinates of the peak voxel in RAI orientation according to the MNI 152 atlas.

(D) RSA results: single-subject and group correlations between RSs. The table contains the results from Representational Similarity Analysis (RSA). The single-subject correlation values are reported, along with the group-level correlation (i.e. obtained from the averaged RSs across subjects) and with the *p*-values resulting from the Mantel test. Kinematic = *kinematic synergy* model; EMG = *muscle synergy* model; ID= *Individual Digit* model. The accuracy values reported in red are not significant according to the Mantel test (10,000 iterations).

(E) RSA results: single-subject and group correlations between behavioral and fMRI RSs. The table contains the results from Representational Similarity Analysis (RSA) between each behavioral model and fMRI data. The single-subject correlation values are reported, along with the group-level correlation (i.e. obtained from the averaged Representational Spaces – RSs – across subjects) and with the *p*-values resulting from the Mantel test. Kinematic = *kinematic synergy* model; EMG = *muscle synergy* model; ID= *Individual Digit* model. The accuracy values reported in red are not significant according to the Mantel test (10,000 iterations).

(F) Goodness of fit between original and decoded hand postures. Average goodness-of-fit ($R^2$) values and Standard Deviations (STD) between original and reconstructed sets of joint angles related to specific hand postures across all subjects. The decoding procedure allowed us to obtain the set of synergies related to each grasping motor acts directly from fMRI activity, thus to reconstruct the different hand postures across participants.

(G) Rank accuracy values between original and decoded hand postures. The table reports the rank accuracy values for the discrimination between the original and decoded sets of joint angles related to specific hand postures across all subjects. The decoding procedure allowed us to obtain the set of synergies related to each grasping motor acts directly from fMRI activity, thus to reconstruct the different hand postures across participants.

(H) Encoding accuracy values for the picture-related brain activity. To assess to what extent the visual presentation of objects might have influenced the encoding of BOLD activity in motor regions, the encoding procedure was performed within the same ROI chosen for RSA and posture reconstruction and choosing BOLD activity at five seconds after the visual object presentation as an estimate of brain responses to the visual presentation of target objects. Only the *kinematic synergy* model was used. The chance levels derived from the permutation tests (1000 iterations) are reported, as well as the threshold at *p*=0.05 and the actual *p*-value obtained from the tests against the null distributions of accuracies. The accuracy values reported in red are not significant. The results show that the

procedure is unsuccessful in all subjects and do not account for a confounding role of image-related activity on the posture encoding results.

(I) Encoding accuracy values for kinematic synergies in visual areas. To assess the impact of visual imagery on our results, the encoding procedure was performed within a Region of Interest selected based on the image-related activity (at 5 s after presentation) vs. rest ($q<0.01$, FDR corrected). The encoding of postures (using the *kinematic synergy* model only) was then tested in the voxels forming this ROI. The chance levels derived from the permutation tests (1000 iterations) are reported, as well as the threshold at $p=0.05$ and the actual $p$-value obtained from the tests against the null distributions of accuracies. The accuracy values reported in red are not significant. The results show that the procedure is unsuccessful in seven subjects and therefore it suggests a very limited impact of visual imagery on the posture encoding results.

(J) List of objects. Table displaying the twenty common-use objects (chosen from the 57 in Santello et al., 1998) that were used in this study.

(K) List of marked joints and bones. Complete list of hand joints and bones marked during the optical tracking experiment. Two additional markers were placed on the wrist, for a total of 26 optical markers. (L) EMG features The features that were extracted from the EMG signals are listed above. *Muscle synergies* were quantified through principal components analysis performed across features and EMG electrodes yielding a five-dimensional set of synergies.

(M) Rank accuracy values for 1 to 10 PCs. The table displays the rank accuracy values for the two models derived from kinematic and EMG data, with a number of retained PCs ranging from 1 to 10 (kinematic and EMG synergies) or 1 to 5 (individual digits). The reported values are the accuracy scores averaged across subjects and their SD. Notably, the *individual digit* model could explain only a moderate fraction of the total variance of the kinematic data (mean: 26.59%, range 14.46% to 34.97%). PCA dimensionality reduction was therefore successful as the first five synergies (later used for encoding fMRI activity) could explain a mean variance across subjects of 91.78% in the kinematic data and 72.64% in the EMG data. (N) Group synergies defined by constrained *k*-means The three core kinematic synergies from each participant were grouped across participants with a semi-supervised clustering algorithm (Bilenko et al., 2004). The procedure showed that the first three synergies were highly consistent and had the same rank across almost all subjects (i.e., PC 1 was in the first position in most of the subjects). Overall, 77.78% of the single subject synergies were consistently labeled across subjects. The table represents the three "group synergies" and lists the single-subject synergies that compose each of them.

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

## Appendix: Impact of the number of channels on gesture discrimination from EMG data

It could be hypothesized that the worse performance of the *muscle synergy* model as compared to the alternative *kinematic synergy* or *individual digit* models could be related to its lower dimensionality (five muscles against 26 hand DoFs). Despite previous reports indicate that a reliable gesture discrimination can be achieved from seven (*Weiss and Flanders, 2004*) or fewer muscles (*Ganesh et al., 2007*; *Ahsan et al., 2011*), it is feasible to record a larger number of muscles using advanced EMG devices.

Hence, we verified the impact of the number of EMG channels on the *muscle synergy* model in an independent sample of four healthy young subjects (4 M, age 34 ± 6) using the same experimental paradigm described in the Methods.

EMG data were acquired using a 16-channel Bagnoli 16 EMG recording device (Delsys Inc, Natick, MA, USA). Sixteen electrodes were placed on the hand and forearm using the same placement adopted in our protocol (see Materials and methods and *Figure 1* below) as well as in two distinct protocols with different spatial resolutions (*Bitzer and van der Smagt, 2006*; *Ejaz et al., 2015*). Six runs were acquired, each comprising twenty trials of delayed grasp-to-use motor acts towards visually-presented objects (see Materials and methods).

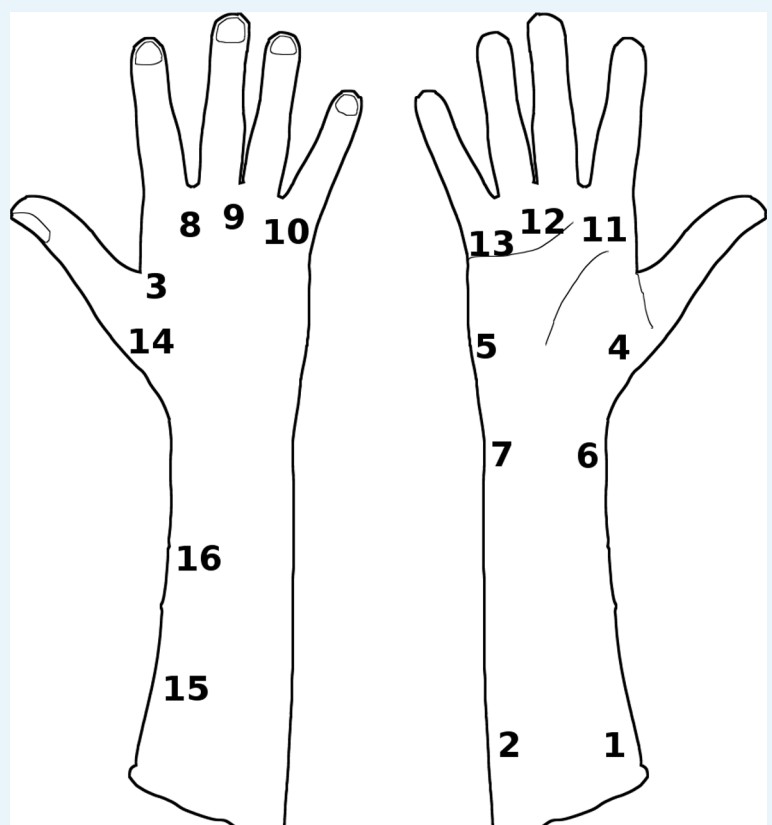

**Appendix figure 1.** Placement of the sixteen electrodes on the right arm.
Four configurations were tested, either with five (1–5, see Materials and methods), ten (1–4, 6–8, 14–16, from *Bitzer and van der Smagt, 2006*), or fourteen channels (from *Ejaz et al., 2015*).

To estimate the impact of the number of EMG recording sites and the preprocessing methods, data were analyzed using two distinct procedures: a mean-based procedure (similarly to *Ejaz et al., 2015*), and a feature-based procedure.

In the mean-based procedure, data from the sixteen EMG channels (acquired at 1000 Hz) were de-trended, rectified, and low-pass filtered (fourth-order Butterworth filter, 40 Hz). The time series from each gesture and channel were later averaged over a 2.5 s time window (2500 time points). From this preprocessing we obtained twenty 16x1 vectors for each run.

In the feature-based procedure, EMG signals were preprocessed and eighty-two features from each channel were extracted as described in the Methods section.

Subsequently, two procedures were developed to uncover the impact of different processing methods and EMG channel configurations. First, we generated all the possible configurations that could be obtained by choosing the channels randomly. Second, we selected three fixed configurations as subsamples of electrodes (displayed in *Figure 1*), according to the Methods in this manuscript (electrodes 1-5) and previous reports that recorded ten (*Bitzer and van der Smagt, 2006*; electrodes 1-4, 6-8, 14-16), or fourteen channels (*Ejaz et al., 2015*; electrodes 1-14).

To allow comparisons across different channel configurations, the EMG matrix (i.e., the averaged EMG activity in the mean-based procedure and the extracted features in the feature-based procedure) was reduced to five dimensions using PCA. Then, both these procedures were assessed with a leave-one-out cross-validation algorithm based on the same rank accuracy measure described in the manuscript.

This additional experiment provides a measure of the quality of each channel configuration: the higher the accuracy, the more informative the configuration. The results are shown in *Figure 2* as the average across combinations and subjects $\pm$ SEM. We tested all configurations that could be obtained by randomly selecting 5 to 16 electrodes (red and blue lines), as well as three fixed configurations according to the setups described above (orange and light blue dots). The red line represents the results using the mean-based procedure, while the blue line depicts the feature-based procedure. The orange and light blue dots represent the results of the three fixed configurations of channels in the two procedures.

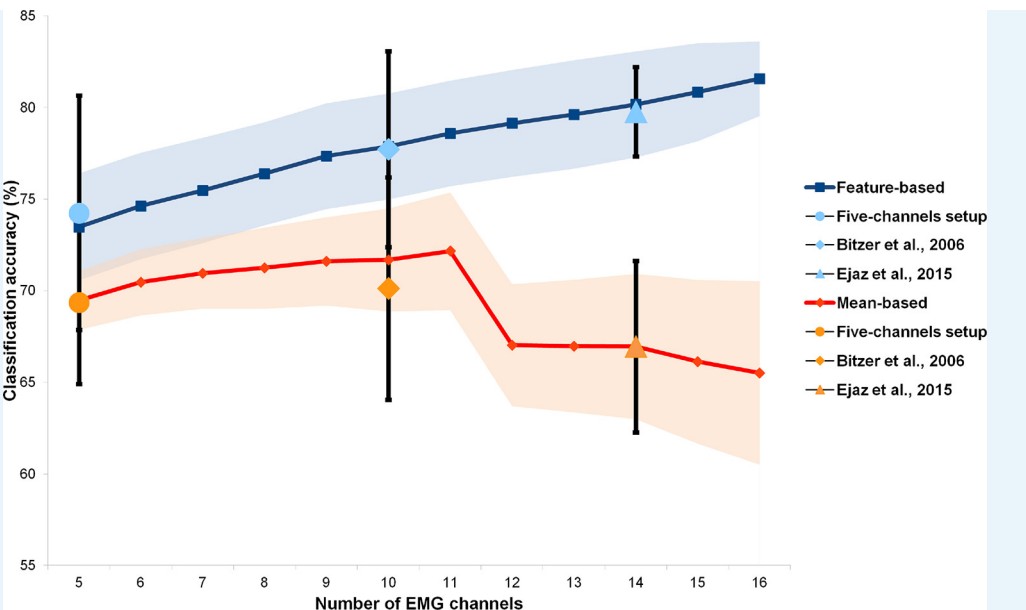

**Appendix figure 2.** Results of the rank accuracy procedure as a function of the number of EMG channels.
The red line shows the accuracy values for random configurations of 5 to 16 electrodes, using the mean-based preprocessing adopted by *Ejaz et al. (2015)*. The orange dots represent the accuracy values for three fixed configurations. The blue line shows the accuracy values for 5 to 16 channels using the feature-based preprocessing (see Materials and methods); the light blue dots show the accuracy for three fixed configurations. Values are reported as mean across subjects ± SEM (error bars and bands).

The results show that, for the feature-based procedure, the accuracy increases as a function of the number of electrodes, reaching a peak with 16 channels (mean ± SEM: 81.6 ± 2%); the mean accuracy across all the possible configurations with five channels is 73.5 ± 2.5%. The accuracy obtained with the setup adopted in our current paper was 74.2 ± 6.4%. For the mean-based procedure described in *Ejaz et al. (2015)*, eleven channels yielded the highest accuracies among all the possible random configurations (value: 72.2 ± 3.2%); accuracy decreased when lower or higher numbers of electrodes were recorded. In these data, the accuracy for the configuration of five channels adopted in our paper was 69.5 ± 1.6%.

Overall, these results indicate that the extraction of features from the EMG signal proves to be a reliable procedure to a discriminate complex hand gestures. In addition, despite the fact that the feature-based approach seems to benefit from EMG recordings with more channels, the gain when raising the number of channel to 16 is low (5.5%). This result, along with the above-chance discrimination achieved when analyzing five channels clearly suggests that the number of muscles recorded in our paper represents the muscle space with a reasonable accuracy. Moreover, feature-based approaches are likely to be better descriptors of more complex gestures (as the ones considered in our study) with respect to the mean signal over time, as hypothesized and discussed in previous reports (*Hudgins et al., 1993*; *Zecca et al., 2002*).

In conclusion, the *muscle synergy* model, even if based on many EMG channels, still underperforms relatively to the models obtained from kinematic data in encoding fMRI responses. For this reason, the worst performance of the *muscle synergy* model is likely to represent an intrinsic limitation of surface EMG signals rather than a flaw of the recordings and analyses performed in our paper.

