## [Decision Letter]

[Editors’ note: a previous version of this study was rejected after peer review, but the authors submitted for reconsideration. The first decision letter after peer review is shown below.]

Thank you for submitting your work entitled "A synergy-based control is encoded in human motor cortical areas" for consideration by *eLife*. Your article has been reviewed by three peer reviewers, and the evaluation has been overseen by Jody Culham as Reviewing Editor and Timothy Behrens as the Senior Editor. Our decision has been reached after consultation between the reviewers.

The following individuals involved in the review of your submission have agreed to reveal their identity: Joern Diedrichsen and Jason Gallivan (reviewers).

Based on these discussions and the individual reviews below, we regret to inform you that we are rejecting the manuscript. As it stands, the manuscript would require substantial revisions, including new data and analyses, to address reservations raised by the reviewers, particularly Reviewer #1. All three reviewers (like the editors in the initial evaluation) saw potential in the approach but their enthusiasm was tempered by a number of concerns. In post-review discussions, even Reviewers #2 and #3, who were largely positive, agreed that the manuscript should not be published unless the major concerns detailed below are addressed. Normally *eLife* tries to avoid making authors go through a gauntlet of revisions if a positive final outcome is uncertain. As such, we are rejecting the current manuscript.

That said, we would be willing to consider a new manuscript that unequivocally addresses the concerns raised. In this case, we would aim to recruit the same reviewers. We must emphasize that, as with any new manuscript, there is no guarantee of publication, especially if you do not make the required changes or the new analyses do not support the conclusions. As such, you may decide instead to submit your manuscript to another journal, in which case we hope the reviewers' comments are helpful.

To be considered for publication in *eLife*, the following changes would be essential (based on the specific reviewer comments detailed below):

1) The authors would need to test their synergy model against a more plausible "muscle model" based on better EMG recordings of more muscles (Reviewer #1, Point #1; though the other two reviewers were in close agreement during post-review discussion). Though the possibility of removing the muscle model from the paper was discussed, the consensus was that this would reduce the impact of the paper.

2) There was a clear consensus among all three reviewers and reviewing editor that the interpretation of principal components and their mapping needs to be unpacked better. One concern is that without any insight as to what the principal components represent, the demonstration that they show a topography is of limited value (Reviewer #2, first point). A second concern is whether the topography really reflects those components or would equally reflect components in other rotated versions of the space (Reviewer #1, Point #3). There may be some potential with the PCA data ameliorate some concerns at the initial review stage and from Reviewer #1 about whether the paper provides a sufficient advance beyond Ejaz et al. (2015). While Ejaz et al. limited their analyses to M1 and S1, the present manuscript shows potentially interesting patterns in other parietal and frontal regions. However, as it stands, the patterns and interpretation so vague that they do not provide any real insight into regional differences.

3) The authors need to clarify their discussion of their PCA-based methods against the RSA based method (as used both in their paper and in Ejaz et al. – see Reviewer #2's comment on RSA and Reviewer #1, Comment #2).

In addition, if the authors choose to resubmit the manuscript in a new form to *eLife*, the other comments of the reviewers should be addressed. In post review discussion, all reviewers agreed with the suggestion that noise ceilings should be reported.

Reviewer #1:

The study "A synergy-based control is encoded in human motor cortical areas" provides an investigation of the MRI patterns associated with the execution of grasp-like hand shapes. The main finding is that the activity patterns of two left –out postures can be better discriminated when using 5 regressors extracted from kinematic synergies than 5 regressors reflecting the unsigned displacement of individual fingers or 5 regressors picked from features of EMG recording of 5 different hand muscles. The conclusion of the study are largely overlapping with that of an earlier paper from our lab (Ejaz et al., 2015), but the study adds a number of interesting extra aspects to this line of work, including testing the generalisation of the model to new postures and the investigation of the spatial arrangement of these synergies onto the cortical sheet (but see point 3). The current version of the paper, however, has a number of weaknesses that certainly would need addressing.

1) The alternative models (individual fingers and muscle) give the appearance of straw men. The individual finger uses the L1 norm of movement of each finger – so in contrast to the kinematic synergy model, it does not distinguish between finger flexion and finger extension. This decision appears to be somewhat arbitrary. So, it leaves the reader with the question of whether there is something special about taking the absolute value, or about the specific rotation of these 5 factors in representational space. A more convincing line of investigation would be to try to use optimisation to rotated the 5 linear factors in the kinematic space as to get the best possible decoding performance, and then test the closeness of this solution with the one provided by the kinematic synergy model.

Matters are worse with the muscle model. The authors recorded 5 muscles only, despite the fact that in our experience it is feasible to get 14 or more distinct signals from hand muscles from surface electrodes (Ejaz et al., 2015). These may not always reflect individual muscles, but that is hardly important if we only want to obtain a representative picture of the space of muscle activity. The extracted features from the EMG signals appear obscure to me; and ultimately, the ability to distinguish between postures based on this data is very bad, indicating that most of these numbers reflect noise. Given these large a-priori differences in the quality of the models, I think any subsequent difference in how well fMRI activity patterns can be predicted become utterly unconvincing. So I think the authors need to work harder on trying to equate the reliability of their models (for a possible method, see Ejaz et al. (2015), supplementary methods).

2) While I think that some of the techniques used in the paper are interesting and promising, I believe that they are currently not going beyond the RSA analysis presented in Ejaz et al. Unfortunately, in the discussion the authors misconstrue the previous evidence and perpetuate some misunderstandings that are all too common in the synergy field. For example the authors state that "these findings, however, provide no clue regarding the extent to which the brain may control the hand using functional modules" and "their model (hand usage model) was therefore similar to the individual digit model adopted in the present study", showing that they clearly do not appreciate the tight connection between RSA and the methods chosen here.

It is important to point out that the matrix of pairwise distances contains the same information as the covariance matrix between experimental conditions (and can easily be transformed into it). Extracting PCA factors from this covariance matrix provides a distribution of the same statistical quantity, only that it throws away a certain proportion of the information (and by just considering the principle vectors also disregards the relative importance of the factors). Therefore RSA and extraction of principal components from the covariance reflect highly related information – and I do not think that the authors have made any convincing case that anything can be learned from the PCA approach that cannot be learned from looking at the whole space. This boils down to the key question in the synergy field of whether there is something special about the principal vectors (or synergies) themselves, or just about the representational space they describe. With the current evidence, the difference between this paper and Ejaz et al. is purely superficial and methodological, but not conceptual.

Similarly, our hand use model is not equivalent to the individual finger model used in this paper, but is much closer to the kinematic synergy model (we present a single-finger model in the multi-finger experiment, which is much inferior to the natural statistics model). The authors choose to extract synergies by taking 20 postures that serve as ad-hoc samples of the natural statistics of movement. In our paper we chose to use data sets that are representative samples of the natural statistics of movement. Furthermore, we use the whole covariance matrix of the data to compare to the brain activity patterns, not just the first 5 factors.

Indeed, in the analysis of the multi-finger experiment in that paper, we started with very similar methods employed by the authors here, but ultimately decided to present only the RSA methods, as we believe that they show the main point of the similarity of representational space more concisely, than the extraction of some arbitrary number of main factors, which then serve as descriptors of the same space.

3) The mapping of synergies on the cortical sheet is an interesting addition and provides a real potential argument that the kinematic synergies are more that statistical descriptors of representation space, but that the factors themselves have special status. The problem, however, is that currently the one single mapping is not evaluated against many other possible mappings. Thus the authors have not shown that there is something special about the synergies extracted. For this, one would need to a) develop a measure of the "topological orderliness" of the mapping and b) compare the synergy map systematically against an exhaustive set of alternative rotations in the same space (again using optimisation). We actually attempted this analysis on our multi-finger experiment, but preliminary results were not terribly encouraging, as there seem to be rotations of these factors in the same rotational space which gave similarly orderly mappings. If the authors could show in a stringent and convincing fashion that the particular rotation chosen here is more orderly organised than any other possible rotation in the 3-dimensional space (or even conclude after careful evaluation that this is not the case), I think the paper would really increase in quality. Without such analysis Figure 2 remains merely suggestive and anecdotal and the claims not substantiated.

Reviewer #2:

This is excellent work. The study is well thought-out and executed, the paper is clearly written and the analyses are rigorous and appear to have been conducted with care. I suspect that the experimental question asked and results obtained will be of general interest to the sensorimotor research community and the authors do a good job of integrating and motivating their study based on what is currently known.

While I do not have any significant concerns about the work, there are a few points, summarized below, that I think should be considered in a revision.

1) Given the data-driven nature of the analyses used, I found some interpretation of the top 3 principal components, and their relation to the topography noted, lacking. Ultimately the insight provided by PCA in neurophysiology rests on being able to directly link the components to neural activity. While I agree that there is some general map of the components in sensorimotor cortex, their organization has no interpretation. Some interpretation of the components (PCs 1-3) and how they relate to cortex organization might be informative on this front. Otherwise, simply saying PCs are mapped onto cortex is fairly impenetrable for the reader.

2) I found the intermixing of results material into the Discussion section a bit disruptive (e.g., inclusion of Figure 5 and visual control analyses). I think that results material should be described and motivated in the Results section.

3) It was unclear to me why, after measuring from five muscles, and thus obtaining five measures (i.e., the same number of components in the synergy and individual digit models), the data was reduced through PCA and then up-sampled again (through cross-validation methods) to achieve 5 components. This should be fully explained, as no such manipulation was done to the individual digit data.

4) It might be interesting in the supplement to show the results of RSA/MDS for the other models (EMG and individual digit), allowing the reader to make comparisons between all 3 models.

5) I was initially confused in the Discussion why the authors referred to brain areas that were not apparent in the group maps shown in Figure 1 (e.g., ventral premotor cortex). This became apparent, however, after I viewed the actual source data on the MNI-152 brain, as 2-3 subjects show overlap in some of these areas. In any case, the authors should only use the text to refer to what is actually shown in the paper, to avoid such confusion.

6) In the Discussion, I was hoping for some discussion of the bilaterality of the effects observed, which are interesting. I would suggest adding this in a revision.

7) In addition to the visual control used in the paper (i.e., analysis of visual stimulation evoked time points in the RSA mask), I was thinking that an equally good control to show the selectivity of effects to sensorimotor cortex would be to localize much of visual cortex (e.g., based on visual stimulation response vs. rest) and then perform the exact same encoding analyses on those voxels. Visual cortex is well known to be involved in imagery-a key component of the experimental task-and to see how the kinematic models performs in that area would be of interest. If it does fairly well, it would have some significant bearing on what is actually being measured in sensorimotor cortex as well as its underlying organization.

Reviewer #3:

This study investigates whether and to what extent kinematics or muscle synergies are represented in the human motor system during grasping toward virtual objects. To this aim, authors measured hand kinematics and electromiography (EMG) signals and used these information to create a kinematic synergy model, an individual-digit model and a muscle synergy model. By computing correlations between each of these models and brain activity during grasping of virtual objects, they found that the kinematic synergy model explained better the fMRI data than the other models in various motor areas. The authors concluded that the control of hand postures in the brain is based on kinematics synergies.

This study addressed a very interesting question in the motor control field by using the state of the art fMRI analyses and combining various measurements like kinematics, EMG and fMRI data. I find the results and the conclusions of this study highly relevant for the understanding of the motor control system and for the advance of neuroprosthetics.

[Editors’ note: what now follows is the decision letter after the authors submitted for further consideration.]

Thank you for resubmitting your work entitled "A synergy-based control is encoded in human motor cortical areas" for further consideration at *eLife*. Your revised article has been favorably evaluated by Timothy Behrens (Senior editor), Reviewing editor Jody Culham, and by Joern Diedrichsen, one of the original reviewers.

The external reviewer has devoted a considerable amount of time to re-examining the revision and has had detailed conversations with the editors to make the point clear. We are now at the following position: Two of the original reviewers thought the manuscript was strong and we believe all of their concerns have been addressed. We also think the new manuscript is improved and remains of substantial interest but have several remaining concerns (see list below). Out of these concerns there is one point that is really essential and has a major impact on the interpretation of the study. Let us be clear, we think the manuscript is of interest and we intend to publish it, but we do not think that your current analyses support that claim that the synergies reflect actual cortical codes. The figure where you try to make this point (Figure 2) is not subjected to the correct tests to make this point (as you concede in your response to Reviewer 1, point 3 in your response).

*eLife* tries not to subject authors to endless rounds of revision, but we would like to give you one more opportunity to revise the paper to take into account the new comments. Essentially we are asking you to either (a) to perform a proper analysis which shows that the synergies do a better job of representing the cortical patterns than other rotations of the covariance space, or (b) to be clear in the manuscript that this claim is not supported and that the claim that you can support is that there covariance space as a whole is well-represented.

There is also confusion as to how to interpret the text around this critical analysis, which you describe is "Resistant" to rotations in the covariance space. It is clear that the analysis is not invariant to such rotations as described in point 11 below. When we read the text surrounding this description, we read the exact opposite interpretation from each other. One of us thought you were saying the R^2^ value was invariant to rotations and therefore resistant. Another thought you were saying that the R^2^value was NOT invariant, and therefore the overall analysis was resistant. The R^2^values change with rotations, and it is important that manuscript is clear about this. However, it is also clear that this point is not sufficient to demonstrate that the particular eignenvectors that you arrive at via the PCA are the ones that are encoded topologically in cortex. To make this point, you would need to compare them to other rotations of covariance space as described above, and in the previous review.

For clarity for a broad readership, if indeed you can show that the three principal components are better than other rotations, the Reviewing Editor thought that some of your wording in the reply to reviewers would be helpful to include in the manuscript itself ("However, a remarkable body of literature indicate that the highest-ranked kinematic PCs correspond to strictly coded grasping primitives (Santello et al., 1998; Gentner and Classen, 2006; Ingram et al., 2008; Thakur et al., 2008; Gentner et al., 2010; Overduin et al., 2012), see Santello et al., 2013 for review). […] In our study, we examined the first two PCs, which were highly consistent with the literature, along with a third one representing a movement of flexion and thumb opposition (as to grasp a dish or a platter)."

Reviewer #1:

Upon reading the revision, I think the authors have addressed some points raised in the original critique, whereas in other areas I found the response and the changes to the manuscript not satisfactory. This may be partly due to a strong philosophical difference regarding what synergies are and how to interpret the evidence – where I seem to fundamentally disagree with the authors. But I agree that the paper provides an interesting additional and alternative viewpoint to Ejaz et al. (2015) – so I do not want to stay in the way and would recommend publication after a number of clarifications and corrections have been made.

Overall I still found the methods and analysis presented in the paper still somewhat obscure and relatively hard to follow. In interest of clarity and transparency, I would therefore urge the authors to clarify the remaining points in the manuscript. It is the policy of *eLife* to not restrict length or supplimentary materials to allow the presentation of self-contained papers, and the authors should really try to be as clear as possible.

1) "Individual-digit model, based on a somatotopic criterion (Kirsch et al. 2014)" remains still as obscure as it was before. I urge you to clarify here in the Introduction. There is no notion of somatotopy in the individual-digit model as far as I can see. Somatotopy implies that it matters that the middle finger is closer to the ring than to the pinkie finger. The Individual-finger model treats all finger movements equally and independently – so it is not "somatotopic".

2) Results section, first paragraph: I found the analysis provided on the additional 4 subjects interesting and thank you for the additional clarification, but would ask you for two things: a) When using temporal averaging of the EMG signal (mean-based EMG analysis), you should use your dimensionality reduction to 5 PCA, as you did for the feature based analysis. This way we can clearly see that your temporal features, and not the dimensionality reduction provide the critical difference between the red and blue curve. Note also that you did not replicate exactly the analysis performed in Ejaz et al. (2011), as you skipped the critical prewhitening step. It is not clear whether this analysis would be sensible here, as your gestures are a ad-hoc sample from the natural statistics, not an equally-spaced sample of possible finger movements b) this analysis should be included as supplementary material and cited from the main text.

3) Supplementary file 15: I think the table should be supplemented by a one-sentence description for each feature that is detailed enough to be able to calculate these features without going onto a wild-goose chase in the cited papers. I urge the authors to start with a clear definition of symbols and then give a concise and unambiguous formula.

4) Section “A challenge to individual digit correction representations? The functional topography of hand synergies”: I find this section on functional topography overstated and do believe it requires a major change in tone. Your data shows that there is "some" topological organization of the first three synergies, not a "strict" one. Furthermore, some somatotopic clustering can also be shown for individual fingers or – most like for other rotations of the synergy vectors, and you have not provided a quantitative comparison with other possible organizations (see point 11).

5) Section “Limitations and methodological considerations”: The limitation section discusses relatively minor points. Two important weakness should be added: a) the point that while some clustered representation was shown in sensory-motor regions, you did not convincingly show that this specific set of synergies is more clustered than other rotation of the same vectors b) that in comparing the different models, the EMG-model had much less ability to discriminate different gestures and that the disadvantage of the muscle model may simply reflect noise levels on your measurement. These are important limitations that should be pointed out.

6) Paragraph two, “Models validation”: Please clarify in the text how the labels of the test set where shuffled. Specifically, if your test set contained 4 repetitions of each of the 20 gestures, did you shuffle the labels of all 80 trials completely randomly, or did you keep the 4 trials for the same gesture together and just give them together a new label (or equivalently shuffle the labels after averaging over the 4 trials)? This difference has important consequences for the variance of your reshuffling statistics.

7) “Every voxel had a score ranging from 0 (if the voxel was never used) to a possible maximum of 380 (if the two left-out patterns could be predicted, for that voxel, in all the 190 iterations).” Please explain this statement better. Do you mean to say the score was the number of times the voxel was included in the 1000 voxels AND got a specific gesture correct?

8) Section “Assessment of the accuracy of the encoding analysis”: Please clearly point out in the text that the weights were randomly shuffled within each column. Please also point out explicitly (I assume that this is true) that the new "PCA"s were now not orthogonal to each other anymore.

9) Now that I think that I understand what the single subject maps are, I think the group-level maps also needs some more explanation. The score for each voxel varied between 0 and 380 (as stated above). For each subject, which value was then considered as "successful"? Why was it called a "probability map"? Probability of what?

10) Section “Cortial mapping of the three group synergies”: I disagree that using R^2^ as a goodness of fit for each individual synergy makes the results invariant to rotations in synergy space. It does not. Maybe we fundamentally misunderstand each other, so I will make my point more concrete. Say, you have 2 "synergies" of 5 elements X1 and X2 and a 5-element data series Y.

x1=[-2 -1 0 1 2]';x2=[1 0 -2 0 1]';

Y=[-2 -2 1 2 2]';

Then the R2-values of each of the columns of X can be calculated as

R2_1 = Y'*x1*inv(x1'*x1)*x1'*Y/(Y'*Y) = 0.847

R2_2 = Y'*x2*inv(x2'*x2)*x2'*Y/(Y'*Y) = 0.039

Now I rotate

R=[cos(0.9) sin(0.9);-sin(0.9) cos(0.9)];

Z=[x1 x2]*R;z1 = Z(:,1);z2 = Z(:,2);

Now the individual R^2^-values are changed, and hence a mapwise evaluation criterion would also be changed.

R2_1 = Y'*z1*inv(z1'*z1)*z1'*Y/(Y'*Y) = 0.6351

R2_2 = Y'*z2*inv(z2'*z2)*z2'*Y/(Y'*Y) = 0.4629

I hope that clarifies my point and why I think a) the sentence stating that individual R^2^ values are rotation invariant should be removed and b) any claims regarding a special organisation on the cortex should be made weaker – for stronger claims you would need to compare the C-metric (which I think would fit this purpose) across many different rotations of the same vectors or ways of picking encoding vectors from the 20-dimensional space.

---

## [Author Response]

[Editors’ note: the author responses to the first round of peer review follow.]

*In addition, if the authors choose to resubmit the manuscript in a new form to eLife, the other comments of the reviewers should be addressed. In post review discussion, all reviewers agreed with the suggestion that noise ceilings should be reported*.

*Reviewer #1:*

*The study "A synergy-based control is encoded in human motor cortical areas" provides an investigation of the MRI patterns associated with the execution of grasp-like hand shapes. The main finding is that the activity patterns of two left -out postures can be better discriminated when using 5 regressors extracted from kinematic synergies than 5 regressors reflecting the unsigned displacement of individual fingers or 5 regressors picked from features of EMG recording of 5 different hand muscles. The conclusion of the study are largely overlapping with that of an earlier paper from our lab (Ejaz* et al.*, 2015), but the study adds a number of interesting extra aspects to this line of work, including testing the generalisation of the model to new postures and the investigation of the spatial arrangement of these synergies onto the cortical sheet (but see point 3). The current version of the paper, however, has a number of weaknesses that certainly would need addressing.*

*1) The alternative models (individual fingers and muscle) give the appearance of straw men. The individual finger uses the L1 norm of movement of each finger* – *so in contrast to the kinematic synergy model, it does not distinguish between finger flexion and finger extension. This decision appears to be somewhat arbitrary. So, it leaves the reader with the question of whether there is something special about taking the absolute value, or about the specific rotation of these 5 factors in representational space. A more convincing line of investigation would be to try to use optimisation to rotated the 5 linear factors in the kinematic space as to get the best possible decoding performance, and then test the closeness of this solution with the one provided by the kinematic synergy model.*

The Reviewer is absolutely correct in stating that finger flexion and extension fully deserve to be kept as distinct as possible. As a matter of fact, our *individual digit* model was actually obtained by summing the individual joint angles for each digit, though in the description in the text we erroneously reported that it was the L_1_-norm of each digit. Consequently, our original individual digit model already distinguished positive and negative joint angles (with respect to the resting posture), thus preserving the difference between flexion and extension. We apologize for this erroneously reported piece of information.

*Matters are worse with the muscle model. The authors recorded 5 muscles only, despite the fact that in our experience it is feasible to get 14 or more distinct signals from hand muscles from surface electrodes (Ejaz* et al.*, 2015). These may not always reflect individual muscles, but that is hardly important if we only want to obtain a representative picture of the space of muscle activity.*

Previous reports indicate that a reliable gesture discrimination can be achieved from seven (Weiss & Flanders, 2004; Shyu et al., 2002) or fewer muscles (Ganesh et al., 2007; Ahsan et al., 2011) and that, since EMG data are notoriously prone to artifacts such as cross-talk or amplitude cancellation (i.e., the masking of small or deep muscles by bigger or superficial ones), it is fairly important to place the electrodes as close as possible to individual muscles. In our study, both the number of recorded muscles and the analyses performed were able to guarantee a reliable discrimination between gestures, as indicated by the rank accuracy measure, which was well above the chance level for each subject ([Supplementary-material SD4-data] and Figure 6).

In most advanced EMG devices, the number of acquisition sites on the forearm can raise dramatically up to 128 or 192 (Gazzoni et al., 2014; Muceli et al., 2014). However, due to the above mentioned cross-talk, the effect of the potential benefit that could derive from an increased number of EMG electrodes is still much debated: indeed multi-channel setups show a high collinearity, since many channels inevitably record the same muscles. As a matter of fact, a recent work that directly compared multiple channel setups – using either 6, 8, 16 or 192 channels – and extracted five synergies, found no significant differences among the 6, 16, 128 or 192 channel configurations (Muceli et al., 2014).

Nevertheless, in order to explore directly the Reviewer’s critique, we tested an independent sample of four healthy young subjects who performed the very same task used in our current paper to verify the impact of the number of EMG channels on the ‘muscle’ model. To this aim, for each subject we acquired six runs, each comprising twenty trials of delayed grasp-to-use motor acts towards visually-presented objects. Data were acquired using a Bagnoli 16 EMG recording device (Delsys Inc, Natick, MA, USA). Sixteen electrodes were placed on the hand and forearm using the same placement adopted in our protocol (see revised Methods and Appendix figure 1) as well as in three distinct protocols with different spatial resolutions (Bitzer and van der Smagt, 2006; Ganesh et al., 2007; Ejaz et al., 2015).

To obtain an overall estimation of the impact of the number of EMG recording sites and different preprocessing steps, data were analyzed using two distinct procedures: the first suggested by the Reviewer (Ejaz et al., 2015), and the other one based on feature extraction, as described in our Methods. Both preprocessing procedures were assessed with a leave-one-out cross-validation algorithm.

Using the first procedure (Ejaz et al., 2015), data from the sixteen EMG channels (acquired at 1,000 Hz) were de-trended, rectified, and low-pass filtered (fourth-order Butterworth filter, 40 Hz). The time series from each gesture and channel were later averaged over a 2.5 seconds time window (2,500 time points). We obtained twenty 16x1 vectors for each run. Then, we tested the discriminability of each individual movement (*probe* element) from each run against all the movement vectors averaged across the five remaining runs (*rest* dataset). This was performed with a leave-one-out rank accuracy procedure (Mitchell et al., 2004) in which the similarity between the *probe* element and all the vectors in the *rest* dataset were compared using the Mahalanobis distance. If the distance between the *probe* element and the vector from the *rest* dataset, which represents the same gesture, is lower than the distances between the *probe* vector and the other elements of the *rest* dataset, one may state that the element can be discriminated. The accuracies were tested against null distributions of 10,000 values, generated shuffling the labels in the *rest* dataset. This rank accuracy procedure provides a measure of the quality of each individual channel configuration: the higher the accuracy, the more informative the configuration. Since we wanted to estimate the discriminability as a function of the number of channels, we performed two tests, whose results are shown in Appendix figure 2.

First, we generated all the possible configurations that could be obtained by choosing the channels randomly. The rank accuracy procedure was performed for each of these channel configurations. The result of this procedure is shown in Appendix figure 2 (red line).

Second, we selected four subsamples of electrodes (displayed in Appendix figure 1; electrodes 1-4, 6-8, 14-16), or fourteen channels (Ejaz et al., 2015; electrodes 1-14). Results for each configuration, averaged across subjects, are represented by the orange dots in Appendix figure 2.

In the feature-based procedure, we computed, for each trial, eighty-two features from each channel, as described in the Methods section of our manuscript. Each individual gesture was then described as a combination of five PCs (muscle synergies), extracted from the features pooled across channels. Subsequently, a machine learning procedure based on a rank accuracy measure was employed to test to what extent the gestures could be discriminated based on the five muscle synergies (see Methods). As done with the procedure previously described, we tested all configurations that could be obtained by randomly selecting 1 to 16 electrodes (Appendix figure 2, blue line), as well as four subsamples according to the setups described above (Appendix figure 2, light blue dots).

The results show that, for the feature-based procedure, the accuracy increases as a function of the number of electrodes, reaching a peak with 16 channels (mean ± SEM: 81.6 ± 2%); the mean accuracy across all the possible configurations with five channels is 73.5 ± 2.5%. The accuracy obtained with the setup adopted in our current paper was 74.2 ± 6.4%, while the electrode placement described in Ejaz et al. (2015) led to an accuracy value of 79.8 ± 2.4%. For the procedure described in Ejaz et al. (2015), five channels yielded the highest accuracies among all the possible random configurations (value: 65.9 ± 2.1%); accuracy decreased when lower or higher numbers of electrodes were recorded. In these data, the accuracy for the configuration of five channels adopted in our paper was 69 ± 1.6%, higher than the one adopted in Ejaz et al. (59.6 ± 2.4%).

In summary, these results indicate that:

1) The extraction of features from the EMG signal obtained using the methodological approach adopted in our paper leads to a better discrimination of complex hand gestures

2) While the feature-based approach seems to benefit from EMG recordings with more channels, the low gain (5.5%) when raising the number of channel to 16, as well as the above-chance discrimination achieved when analyzing five channels clearly suggests that the number of muscles recorded in our paper represents the muscle space with a reasonable accuracy.

3) The results from the electrode placement adopted by Ejaz et al. (2015) show a lower accuracy, despite its higher dimensionality. A possible explanation for this finding may be related to the different experimental designs: when more complex gestures are considered (as in our design), data are probably more prone to electrode shifts or amplitude cancellation artifacts, whose impact increases with the number of electrodes in the EMG acquisition setup. Moreover, the better results achieved by using the feature-based approach, as compared to the procedure suggested by the Reviewer, may indicate that feature extraction represent a better technique for exploiting the advantages of EMG setups with higher spatial resolution, consistent with the data already present in the literature (Hudgins et al., 1993; Zecca et al., 2002).

Despite the better discrimination accuracy obtained with the feature-based procedure using a higher number of channels, we found that the accuracy values using a ‘muscle’ model based on many EMG channels still underperforms relatively to the models obtained from kinematic data in encoding fMRI responses. Therefore, we conclude that the worst performance of the ‘muscle model' is likely to represent an intrinsic limitation of surface EMG signals rather than a flaw of the recordings and analyses performed in our paper. For the reasons detailed below (Point 1c), we would expect that even high-density EMG data may not guarantee a better fit to brain signals than electrode configurations with a lower dimensionality.

*The extracted features from the EMG signals appear obscure to me; and ultimately, the ability to distinguish between postures based on this data is very bad, indicating that most of these numbers reflect noise. Given these large a-priori differences in the quality of the models, I think any subsequent difference in how well fMRI activity patterns can be predicted become utterly unconvincing. So I think the authors need to work harder on trying to equate the reliability of their models (for a possible method, see Ejaz* et al. *2015, supplementary methods).*

EMG data are complex signals, as they represent indirectly the activity of one or more underlying muscles (Farina et al., 2004; Reaz et al., 2006; Farina et al., 2014). However, a well-consolidated assumption is that the instantaneous value of the EMG signal contains little or no information (Zecca et al., 2002). In addition, during complex movements the temporal structure of the EMG signal varies to a great deal. This makes the most elementary frequency or amplitude-based parameters (e.g. mean absolute value, median frequency) not sufficient for the classification of complex movements: an accurate classification may instead benefit very much from a greater number of signal descriptors (Hudgins et al., 1993). For this reason, the extraction of time-domain or frequency domain features is widely applied when EMG signals are used to classify distinct hand gestures.

Consequently, the most commonly adopted approach relies on the extraction of a wide number of features, which are later analyzed through classification algorithms or neural networks for posture discrimination (Zardoshti-Kermani et al., 1995; Ahsan et al., 2011; Chowdhury et al., 2013). The features chosen in our study are commonly employed for classifying gestures from EMG recordings of hand muscles (for instance, see Zecca et al., 2002; Boostani and Moradi, 2003; Tkach et al., 2010; Kendell et al., 2012; Chowdhury et al., 2013). The matrix containing those features was subsequently reduced in dimensionality, retaining only the most relevant five components, which accounted for 72.64% of the variance (mean across subjects). In addition, the rank accuracy measure showed that the twenty gestures could be successfully discriminated by EMG data in all the subjects (average accuracy value across subjects: 72%; individual subject data are reported in Supplementary File 16).

Nonetheless, even if feature extraction is widely used, we followed the Reviewer’s suggestion and adopted the elegant RSA-based procedure performed by Ejaz and colleagues (Ejaz et al., 2015, Supplementary Methods). EMG signals from our five-channels setup were resampled at 1,000 Hz, de-trended, rectified and low-pass filtered with a fourth-order Butterworth filter at 40 Hz. The time series from each gesture and channel were later averaged over a 2.5 seconds time window (2,500 time points). As a result, we described each gesture using a 5x1 vector. A representational space (RS) was then obtained from the pairwise comparisons of all gestures, performed using the Mahalanobis distance. That RS turned out to be highly similar to our muscle model (Pearson’s *r*: 0.78, *p*<0.0001). This suggests that feature extraction is able to provide a reliable description of the muscle space and that a different data analysis would not have had a significant impact on the model we derived from EMG data.

We believe that two additional details should be pointed out. First, we described our EMG signals using a high number of features, subsequently reduced to the five most relevant dimensions with PCA. Ejaz et al., (2015) described the EMG signal for each movement exclusively through its average over time. While this is certainly suited for simpler single or multi-digit movements as the ones examined in their study, many pieces of evidence in the literature indicate that complex hand actions, including grasping movements, require a more detailed description (Hudgins & Parker, 1993; see Zecca et al., 2002 for review). Second, an analysis limited to a descriptive RSA-based approach would not have allowed us to perform the rank accuracy procedures to evaluate the discrimination of individual postures from the EMG data, which was an important and innovative aim of our study.

*2) While I think that some of the techniques used in the paper are interesting and promising, I believe that they are currently not going beyond the RSA analysis presented in Ejaz* et al. *Unfortunately, in the discussion the authors misconstrue the previous evidence and perpetuate some misunderstandings that are all too common in the synergy field. For example the authors state that "these findings, however, provide no clue regarding the extent to which the brain may control the hand using functional modules" and "their model (hand usage model) was therefore similar to the individual digit model adopted in the present study", showing that they clearly do not appreciate the tight connection between RSA and the methods chosen here.*

*It is important to point out that the matrix of pairwise distances contains the same information as the covariance matrix between experimental conditions (and can easily be transformed into it). Extracting PCA factors from this covariance matrix provides a distribution of the same statistical quantity, only that it throws away a certain proportion of the information (and by just considering the principle vectors also disregards the relative importance of the factors). Therefore RSA and extraction of principal components from the covariance reflect highly related information – and I do not think that the authors have made any convincing case that anything can be learned from the PCA approach that cannot be learned from looking at the whole space. This boils down to the key question in the synergy field of whether there is something special about the principal vectors (or synergies) themselves, or just about the representational space they describe. With the current evidence, the difference between this paper and Ejaz* et al. *is purely superficial and methodological, but not conceptual.*

The main finding of our paper is that synergies can be encoded as such in the human brain, and that they may represent functional modules through which the brain controls the hand to achieve a wide variety of flexible and stable postures. The extraction of Principal Components from the covariance matrix is therefore a cornerstone of our current work, since we were interested in demonstrating that a model based on synergies (or kinematic PCs) can predict brain activity in motor areas.

The Reviewer is right in pointing out that “*the matrix of pairwise distances contains the same information as the covariance matrix between experimental conditions (and can easily be transformed into it)*” and that “*RSA and extraction of principal components from the covariance reflect highly related information*”. The high fraction of variance explained by the *kinematic synergy* model reflects a high degree of similarity between the whole space and its five principal dimensions. However, we can hardly see this as a limitation of our approach. On the contrary, we observed that a small number of PCs can describe complex postural configurations losing only a little portion of information, in line with a large body of literature (e.g., Santello et al., 1998, 2002; Gentner and Classen, 2006; Ingram et al., 2008; Thakur et al., 2008; Overduin et al., 2012). For this reason, the high similarity between the whole space and its PCs represents a *success* of the dimensionality reduction procedure and a full validation of the reliability of synergies as descriptions of complex hand postures.

According to the synergy hypothesis, the kinematic Principal Components reflect clearly defined motor primitives and they are not simply quantitative descriptors of hand posture (Santello et al., 1998; Gentner and Classen, 2006; Ingram et al., 2008; Thakur et al., 2008). Following the suggestions made by the other Reviewers as well (see Reviewer #2, Comment #1 and Reviewer #3, Comment #3), we have shown that the first PCs represent elementary grasps (see response below, Methods and Video 1). A direct link between kinematic PCs and motor primitives raises questions about whether hand postures are encoded in the brain as combinations of those elementary grasps, thus confirming modularity principles. To verify this hypothesis, our encoding analyses were purposely designed to predict brain activity as a function of kinematic synergies. In addition, the subsequent decoding procedure tested the reliability of synergies as models to decode hand postures from brain activity.

To conclude, we do appreciate the high quality of the analyses and the innovative results reported in the elegant study by Ejaz et al. (2015). As a matter of fact, we are convinced that our paper complements – rather than simply overlaps with – the findings by the Ejaz et al. (2015) study. Specifically, the complementary nature of our paper is due to significant differences between the objectives of the study by Ejaz et al. (2015) – aimed mainly at providing reliable descriptions of kinematic, muscle and brain activity spaces through RSA – and our work, aimed instead at assessing modularity in hand posture control using machine-learning procedures. Thus, our findings do possess a conceptual novelty as compared to the Ejaz et al. (2015) study.

*Similarly, our hand use model is not equivalent to the individual finger model used in this paper, but is much closer to the kinematic synergy model (we present a single-finger model in the multi-finger experiment, which is much inferior to the natural statistics model). The authors choose to extract synergies by taking 20 postures that serve as ad-hoc samples of the natural statistics of movement. In our paper we chose to use data sets that are representative samples of the natural statistics of movement. Furthermore, we use the whole covariance matrix of the data to compare to the brain activity patterns, not just the first 5 factors.*

*Indeed, in the analysis of the multi-finger experiment in that paper, we started with very similar methods employed by the authors here, but ultimately decided to present only the RSA methods, as we believe that they show the main point of the similarity of representational space more concisely, than the extraction of some arbitrary number of main factors, which then serve as descriptors of the same space.*

We agree with the Reviewer on this point: for descriptive purposes, considering the whole space has surely many benefits. In addition, RSA methods are seemingly more robust, as they allow to compare the postural or functional models and to take into account all the data without performing any reduction.

Nonetheless, as already stated above, we should point out that our paper has a different objective and a different approach. Specifically, we aim at showing that the first PCs of the posture space, in addition to being valid descriptors of that space, do actually modulate brain activity within motor areas, and thus they represent useful modules that can be exploited for prosthetic control and Brain Computer Interface (BCI) applications.

All the Reviewers raised the issue regarding the interpretation of such PCs, as – in the absence of a more detailed description – they could, at first, appear arbitrary. However, a remarkable body of literature indicate that the highest-ranked kinematic PCs correspond to strictly coded grasping primitives (Santello et al., 1998; Gentner and Classen, 2006; Ingram et al., 2008; Thakur et al., 2008; Gentner et al., 2010; Overduin et al., 2012), see Santello et al., 2013 for review). Consistently, in the above reports, a first synergy for grasping was identified, which modulates abduction-adduction and flexion-extension of all the finger joints (both proximal and distal), while a second synergy reflects thumb opposition and flexion-extension of the distal joints only. Maximizing the first synergy leads therefore to a posture resembling a power grasp, while the second one is linked to pinch movements directed towards smaller objects. In our study, we examined the first two PCs, which were highly consistent with the literature, along with a third one representing a movement of flexion and thumb opposition (as to grasp a dish or a platter). Since we thought that a graphical representation of the meaning of synergies could be a useful addition to the interpretation of such PCs, we plotted the time course of hand movements corresponding to these elementary grasps. The plots are 2s-long videos showing three movements from the minimum to the maximum values of PCs 1, 2 and 3, respectively, expressed as sets of twenty-four joint angles averaged across subjects (Video 1).

As indicated above, synergies are meaningful combinations of digit movements: as elementary grasps, they can be considered higher-level representations with respect to single digit or multi-digit movements. For this reason, the kinematic synergy model was tested, in our paper, against an *individual digit* model in which the five digits are considered independently.

In addition, we would like to point out that the hand movement statistics used in the paper by Ejaz et al. (2015) were taken from a previous study (i.e., Ingram et al., 2008) in which the kinematic PCs were later extracted. Notably, the plots from that report show that the first PCs were highly similar to ours (Figure 3 in Ingram et al., 2008).

*3) The mapping of synergies on the cortical sheet is an interesting addition and provides a real potential argument that the kinematic synergies are more that statistical descriptors of representation space, but that the factors themselves have special status. The problem, however, is that currently the one single mapping is not evaluated against many other possible mappings. Thus the authors have not shown that there is something special about the synergies extracted. For this, one would need to a) develop a measure of the "topological orderliness" of the mapping and b) compare the synergy map systematically against an exhaustive set of alternative rotations in the same space (again using optimisation). We actually attempted this analysis on our multi-finger experiment, but preliminary results were not terribly encouraging, as there seem to be rotations of these factors in the same rotational space which gave similarly orderly mappings. If the authors could show in a stringent and convincing fashion that the particular rotation chosen here is more orderly organised than any other possible rotation in the 3-dimensional space (or even conclude after careful evaluation that this is not the case), I think the paper would really increase in quality. Without such analysis Figure 2 remains merely suggestive and anecdotal and the claims not substantiated.*

We completely agree with the Reviewer on this point: the assessment of the “topological orderliness” of hand synergies on the cortex should be an important addition to our paper, as it provides persuasive data towards a real encoding of synergies as the building blocks of hand control in the human brain. Nonetheless, we ought to notice that the demonstration of topological gradients is a major challenge of functional MRI. Since the very first studies which posited a topological organization of some brain regions, e.g., early visual or auditory areas (Sereno et al., 1995; DeYoe et al., 1996; Formisano et al., 2003), most of the reports discussed this type of organization just in a descriptive way, displayed the cortical activations in each experimental subject and described each map individually (DeYoe et al., 1996; Formisano et al., 2003). This approach is typically performed in many studies in which portions of cortex are mapped, often using ultra-high-field fMRI (Formisano et al., 2003; Olman et al., 2010; Sanchez-Panchuelo et al., 2014).

The papers which provide methods to measure topography are very scarce. For instance, Engel et al. (1997) correlated the visual field mapped by V1 as a function of the distance from the occipital pole, obtaining a value which represents the topographical organization of eccentricity maps.

In the present study, we used methods developed by other authors to analyze electrocortical recordings (Yarrow et al., 2014), to measure the orderly mapping of synergies across the regions recruited by the encoding procedure. The method is based on the comparison between the map space (the physical distance between voxels) to the feature space (the distance between voxels based on their information content). The two spaces can be correlated to each other, estimating an index (C parameter) which is, in fact, conceptually similar to the correlation coefficient between two RSs (Yarrow et al., 2014). The C coefficient reflects the similarity between the arrangement of voxels in space and the arrangement of their information content, high values indicate that voxels that contain similar information are also spatially close. This closeness can be considered as a demonstration of a topographical organization, as the distance between the content of voxels is reflected by their physical distance.

Here, we measured the physical distance (voxel space) as the distance (i.e., standardized Euclidean distance) between each voxel. The feature space was computed measuring the distance (i.e., standardized Euclidean distance) between the three synergy weights, as defined by their R^2^, for each voxel. This analysis yielded a significant C coefficient (C=0.192; *p*-value=0.0383), denoting high similarity between the feature space and the voxel space; this supports the existence of a topographical organization of synergies across the cortical surface.

We thank again the Reviewer for this comment, which gave us the opportunity to assess the synergy-based organization using a procedure that had not yet been applied to fMRI data.

The synergy map displayed in Figure 2 was obtained plotting the R^2^ coefficient between each synergy with the fMRI data. This is particularly important because, as R^2^ represents the goodness-of-fit of the model, is resistant to the possible rotations of the three components in the same space, while the β (the angular coefficient of the regression) can be altered by the possible alternate configurations (e.g., rotations) of the principal components. The choice of the goodness-of-fit better suits our task, as it can be divided into two sub-tasks with opposite sign (i.e., performing the grasping movement and returning back with the hand in its original position), which cannot be disentangled due to the low temporal resolution of fMRI.

Overall, our results show that each synergy coefficient, being expressed as a measure of the goodness-of-fit instead of an individual β value, represents a unique measure which can be mapped onto the cortical surface without the need for testing it against alternate rotated configurations. In addition, the test performed shows that the feature space is strongly connected to the voxel space, providing the demonstration of the topological arrangement of hand synergies in the fronto-parietal cortical network specifically recruited by the encoding procedure.

*Reviewer #2: 1) Given the data-driven nature of the analyses used, I found some interpretation of the top 3 principal components, and their relation to the topography noted, lacking. Ultimately the insight provided by PCA in neurophysiology rests on being able to directly link the components to neural activity. While I agree that there is some general map of the components in sensorimotor cortex, their organization has no interpretation. Some interpretation of the components (PCs 1-3) and how they relate to cortex organization might be informative on this front. Otherwise, simply saying PCs are mapped onto cortex is fairly impenetrable for the reader.*

Since the main aim of our work was to demonstrate that PC-based representation of hand postures are actually encoded as such in human motor cortical areas, it is particularly important to unveil *what* those principal components may represent. The Reviewer’s request of an interpretation of the first PCs gives us an opportunity to clarify the meaning of synergies as the building blocks of hand postures.

Synergies are fundamental motor modules that are controlled as weighted combinations during hand movements. They are defined as Principal Component and usually ranked using the fraction of explained variance; the first paper on kinematic synergies (Santello et al. 1998, Figure 6) identified a first synergy which modulates abduction-adduction and flexion-extension of all the finger joints (both proximal and distal), while a second synergy reflects thumb opposition and flexion-extension of the distal joints only. Consequently, a modulation of the first synergy leads to a posture resembling a power grasp, while the second one is linked to pinch movements directed towards small objects. It is noteworthy that a similar organization was also identified by subsequent independent studies (Gentner and Classen, 2006; Ingram et al., 2008; Thakur et al., 2008; Gentner et al., 2010).

In our study, we showed that the first three kinematic PCs, which account for a major fraction of variance, are topographically arranged on the cortical surface. We added a more detailed interpretation of those PCs in the Discussion. The first two synergies are highly consistent with those reported in the literature, representing power and pinch grasps, respectively, while the third one reflects movements of flexion and thumb opposition (as to grasp a dish or a platter).

Following the Reviewer’s suggestion, we developed a method to represent clearly the kinematic PCs. The video represents the whole course of a hand movement from the minimum to the maximum values of PCs 1, 2 and 3, respectively. Please refer also response to issue #2 by Reviewer #1.

*2) I found the intermixing of results material into the* Discussion section *a bit disruptive (e.g., inclusion of Figure 5 and visual control analyses). I think that results material should be described and motivated in the* Results section.

We agree with this suggestion. The visual control analyses have been moved to a new paragraph, as described in the response to comment #7.

*3) It was unclear to me why, after measuring from five muscles, and thus obtaining five measures (i.e., the same number of components in the synergy and individual digit models), the data was reduced through PCA and then up-sampled again (through cross-validation methods) to achieve 5 components. This should be fully explained, as no such manipulation was done to the individual digit data.*

In accordance with the Reviewer’s suggestion and to strengthen our muscle model, we performed the EMG data analysis avoiding the reduction of the number of channels. Therefore, we were able to achieve a model with five dimensions, fully consistent with the alternative descriptions employed in the study. The features were extracted from the five channels separately; later, the five muscle synergies were obtained as linear combinations of features, by computing PCA on the overall set of features. In this way, we could avoid a great deal of data manipulation, achieving a muscle model built on simpler analyses. Changes have been described and reported in the revised version of the manuscript.

*4) It might be interesting in the supplement to show the results of RSA/MDS for the other models (EMG and individual digit), allowing the reader to make comparisons between all 3 models.*

The MDS procedure was performed on the representational spaces drawn during RSA. RSs from each of the three models (*kinematic synergy, muscle synergy* and *individual digit*) were significantly correlated (*p*<0.0001, corrected for Mantel test, 10,000 iterations). Hence, as their MDS would be very similar, they were omitted for brevity.

*5) I was initially confused in the Discussion why the authors referred to brain areas that were not apparent in the group maps shown in Figure 1 (e.g., ventral premotor cortex). This became apparent, however, after I viewed the actual source data on the MNI-152 brain, as 2-3 subjects show overlap in some of these areas. In any case, the authors should only use the text to refer to what is actually shown in the paper, to avoid such confusion.*

We thank the Reviewer for bringing up this potentially confusing passage which could lead to a misinterpretation of the findings. The analyses were conducted on 3D volume data, then the results were mapped onto the cortical surface using specific software (Brainvisa 4.4). Due to the interpolation which occurs during the mapping process, some clusters more associated to deeper brain regions did not appear on the surface view. However, as those regions (e.g., ventral premotor cortex) were actually recruited by the encoding procedure, they were referred to in the Discussion. We modified the figure legend accordingly.

*6) In the Discussion, I was hoping for some discussion of the bilaterality of the effects observed, which are interesting. I would suggest adding this in a revision.*

We are grateful to the Reviewer for this comment. Indeed, the bilateral recruitment of motor, supplementary motor and parietal areas is quite interesting and, previously, other authors have posited a bilateral engagement of those regions during complex hand movements. This aspect has now been discussed more extensively.

*7) In addition to the visual control used in the paper (i.e., analysis of visual stimulation evoked time points in the RSA mask), I was thinking that an equally good control to show the selectivity of effects to sensorimotor cortex would be to localize much of visual cortex (e.g., based on visual stimulation response vs. rest) and then perform the exact same encoding analyses on those voxels. Visual cortex is well known to be involved in imagery-a key component of the experimental task-and to see how the kinematic models performs in that area would be of interest. If it does fairly well, it would have some significant bearing on what is actually being measured in sensorimotor cortex as well as its underlying organization.*

We agree with the Reviewer that the role of visual cortex in motor imagery is significant and early visual areas likely participate to action preparation, as suggested by a very recent study (Gutteling et al., 2015), in which different actions were decoded based on brain activity in visual regions. To rule out such potential confounds and to verify that synergies modulate brain activity exclusively in sensorimotor regions, we considered visual activations vs. rest (measured five seconds after visual stimulus presentation) and performed the encoding procedure on the resulting mask. This procedure, along with the encoding of visual-related responses in sensorimotor regions originally included in the paper, has been described in a new section (Control analyses) in the Results section. The results show that both early activity in motor regions and activity in extrastriate visual regions do not encode kinematic synergies above-chance, excluding therefore those imagery-related confounds.

[Editors' note: the author responses to the re-review follow.]

*1) "Individual-digit model, based on a somatotopic criterion (Kirsch et al. 2014)" remains still as obscure as it was before. I urge you to clarify here in the Introduction. There is no notion of somatotopy in the individual-digit model as far as I can see. Somatotopy implies that it matters that the middle finger is closer to the ring than to the pinkie finger. The Individual-finger model treats all finger movements equally and independently – so it is not "somatotopic".*

The Reviewer is correct in pointing out that our individual digit model considered only the independence of digit representations, without taking into account their orderly somatotopic arrangement on the cortex. The individual digit model was based on the paper by Kirsch and colleagues (Kirsch et al., 2014), which compared the firing rate of M1 neurons with individual finger kinematics and with their Principal Components. In our paper, we described the individual digit model as somatotopic because a somatotopic arrangement requires the presence of discrete, independent single digit representations at a neuronal level. If the discharge pattern of a M1 neuron is correlated with individual digit kinematics, that neuron can be labeled as digit specific. Thus, following the Reviewer’s comment, we removed the potentially misleading references to somatotopy in the description of the individual digit model.

*2) Results section, first paragraph: I found the analysis provided on the additional 4 subjects interesting and thank you for the additional clarification, but would ask you for two things: a) When using temporal averaging of the EMG signal (mean-based EMG analysis), you should use your dimensionality reduction to 5 PCA, as you did for the feature based analysis. This way we can clearly see that your temporal features, and not the dimensionality reduction provide the critical difference between the red and blue curve. Note also that you did not replicate exactly the analysis performed in Ejaz et al. (2011), as you skipped the critical prewhitening step. It is not clear whether this analysis would be sensible here, as your gestures are a ad-hoc sample from the natural statistics, not an equally-spaced sample of possible finger movements b) this analysis should be included as supplementary material and cited from the main text.*

We thank the Reviewer for his comment and suggestions, which we have integrated into the manuscript. In the mean-based procedure (as we are currently labeling this pipeline, alternative to the ‘feature-based’ approach), we reduced the dimensionality of each model with a PCA, retaining only the first five dimensions, independently from the number of EMG channels initially considered (from 5 to 16). Hence, for each number of channels the data have the same dimensionality, making the comparison more reliable.

Following the request by the Reviewer, we have described the EMG analysis in a self-standing way and we have added this procedure to the article. Also, we have discussed the impact of EMG dimensionality in the Limitations section of the main text of the article (as requested also in Comment #5). The ‘prewhitening’ of EMG signals was not included in our analysis, as we did not find any mention of it in the EMG methods of the paper by Ejaz et al. (2015).

*3) Supplementary file 15: I think the table should be supplemented by a one-sentence description for each feature that is detailed enough to be able to calculate these features without going onto a wild-goose chase in the cited papers. I urge the authors to start with a clear definition of symbols and then give a concise and unambiguous formula.*

We agree with the Reviewer on the importance of such pieces of information to achieve the highest possible clarity and to provide all the necessary information for independent replication of the study. Hence, we have added a description of each feature to the list of features reported in [Supplementary-material SD4-data].

*4) Section “A challenge to individual digit correction representations? The functional topography of hand synergies”: I find this section on functional topography overstated and do believe it requires a major change in tone. Your data shows that there is "some" topological organization of the first three synergies, not a "strict" one. Furthermore, some somatotopic clustering can also be shown for individual fingers or – most like for other rotations of the synergy vectors, and you have not provided a quantitative comparison with other possible organizations (see point 11).*

The Reviewer is right that, though our results indicate a topographical organization of the first three synergies which has been tested against random arrangements of the same synergies across the cortical surface (see Cortical mapping of the three group synergies), this organization should be properly tested against the possible rotations of the synergy vectors. This assessment, though, falls far beyond the aim of the current study, as we discussed in the response to Point #11 and in the Limitations and methodological considerations section of the main text of the manuscript. We have modified the Results accordingly, and indicated the need of additional studies to ascertain the topographical arrangement of motor primitives as indicated by our data.

*5) Section “Limitations and methodological considerations”: The limitation section discusses relatively minor points. Two important weakness should be added: a) the point that while some clustered representation was shown in sensory-motor regions, you did not convincingly show that this specific set of synergies is more clustered than other rotation of the same vectors b) that in comparing the different models, the EMG-model had much less ability to discriminate different gestures and that the disadvantage of the muscle model may simply reflect noise levels on your measurement. These are important limitations that should be pointed out.*

The Reviewer is right in pointing out that the impact of the rotation of synergy vectors deserves a deeper discussion. While the topographical arrangement of synergies across the cortical surface is suggested by our data, its demonstration against the possible rotated configurations of the synergy space needs novel and specific experiments, as we detailed in the Response to Comment #11. This has been discussed in a specific paragraph after the Limitations and Methodological considerations section.

The EMG analyses on the additional subjects suggest that the worse performance of the muscle model is likely to be related to the intrinsic signal and noise levels of surface EMG acquisition, as a much higher number of channels (up to 16) did not bring any significant benefit to gesture discrimination. This has been discussed in the “Limitations and methodological considerations” section.

*6) Paragraph two, “Models validation”: Please clarify in the text how the labels of the test set where shuffled. Specifically, if your test set contained 4 repetitions of each of the 20 gestures, did you shuffle the labels of all 80 trials completely randomly, or did you keep the 4 trials for the same gesture together and just give them together a new label (or equivalently shuffle the labels after averaging over the 4 trials)? This difference has important consequences for the variance of your reshuffling statistics.*

In the permutation test, we adopted the former procedure: the labels were shuffled after averaging the four trials of the rest dataset. For this reason, each permutation matrix still comprised twenty coherent postures. We clarified the description of the rank accuracy procedure in the Methods.

*7) “Every voxel had a score ranging from 0 (if the voxel was never used) to a possible maximum of 380 (if the two left-out patterns could be predicted, for that voxel, in all the 190 iterations).” Please explain this statement better. Do you mean to say the score was the number of times the voxel was included in the 1000 voxels AND got a specific gesture correct?*

Since the accuracy assessment (decoding stage) was performed only on the 1000 voxels with the highest R^2^, the accuracy score for each voxel corresponds to the number of times the voxel was among the 1000 with the highest R^2^ and was successful in the pairwise discrimination of gestures. This was clarified in the text.

*8) Section “Assessment of the accuracy of the encoding analysis”: Please clearly point out in the text that the weights were randomly shuffled within each column. Please also point out explicitly (I assume that this is true) that the new "PCA"s were now not orthogonal to each other anymore.*

Following the Reviewer’s indication, this aspect has been specified in the Methods.

*9) Now that I think that I understand what the single subject maps are, I think the group-level maps also needs some more explanation. The score for each voxel varied between 0 and 380 (as stated above). For each subject, which value was then considered as "successful"? Why was it called a "probability map"? Probability of what?*

The group-level map expresses the probability for each voxel to successfully encode synergy-based representations of hand postures. The single subject maps were first binarized, converting the non-zero scores to 1. The nine binary maps, one for each subject, were then summed, obtaining a group map with values ranging from 0 (for a voxel which was never used) to 9 (for a voxel which was recruited in all subjects). This procedure has been better explained in the Methods and Results.

*10) Section “Cortial mapping of the three group synergies”: I disagree that using R^2^ as a goodness of fit for each individual synergy makes the results invariant to rotations in synergy space. It does not. Maybe we fundamentally misunderstand each other, so I will make my point more concrete. Say, you have 2 "synergies" of 5 elements X1 and X2 and a 5-element data series Y.*

*x1=[-2 -1 0 1 2]';x2=[1 0 -2 0 1]'; Y=[-2 -2 1 2 2]'; Then the R2-values of each of the columns of X can be calculated as R2_1 = Y'*x1*inv(x1'*x1)*x1'*Y/(Y'*Y) = 0.847 R2_2 = Y'*x2*inv(x2'*x2)*x2'*Y/(Y'*Y) = 0.039 Now I rotate R=[cos(0.9) sin(0.9);-sin(0.9) cos(0.9)]; Z=[x1 x2]*R;z1 = Z(:,1);z2 = Z(:,2);*

*Now the individual R^2^-values are changed, and hence a mapwise evaluation criterion would also be changed.*

*R2_1 = Y'*z1*inv(z1'*z1)*z1'*Y/(Y'*Y) = 0.6351 R2_2 = Y'*z2*inv(z2'*z2)*z2'*Y/(Y'*Y) = 0.4629 I hope that clarifies my point and why I think a) the sentence stating that individual R^2^ values are rotation invariant should be removed and b) any claims regarding a special organisation on the cortex should be made weaker – for stronger claims you would need to compare the C-metric (which I think would fit this purpose) across many different rotations of the same vectors or ways of picking encoding vectors from the 20-dimensional space.*

In the encoding of fMRI data, the kinematic synergy model used in the multiple linear regression considered the first five synergies together, thereby fitting the combination of those synergies to brain activity patterns. For this reason, the results from the encoding analysis are resistant to the rotation of the synergy space as a whole and the estimation of the goodness-of-fit of the kinematic synergy model is reliable.

That said, the Reviewer is certainly right in noting that, if each synergy is fitted independently as done for the topographical mapping, its R^2^ coefficient may be affected by the possible rotated configurations of the synergy space. Actually, when mapping the synergies onto the cortical surface, the rotation of the PCs may lead to different R^2^ values and, consequently, will affect the topographical organization, as correctly pointed out by the Reviewer. However, the kinematic PCs that were examined in the present paper were identified and represented as elementary, meaningful grasping primitives (see Results and Video 1). Hence, in our opinion, the topographical organization of those synergies (as reported and assessed with the procedure described in paragraph two section “Cortical mapping of the three group synergies”) may indicate that the cortex encodes functional modules.

While we agree with the Reviewer that a direct assessment of functional topography against other possible rotations is a very interesting issue that requires to be explored to achieve a deeper understanding of the cortical organization of hand movement control, we think that this falls beyond the scope of the current study, in primis for the following main reasons:

– The kinematic synergy model accounts for a portion (40%) of the variance of fMRI data. For this reason, we hypothesize that different rotations of the synergy space can actually be encoded in the brain, as well as additional pieces of information (force, temporal patterns of movements, action goals). Without a better model able to explain a larger amount of fMRI variance, the assessment of different rotations of the synergy space may have a limited validity. We discussed this aspect in the Beyond synergies paragraph.

– An encoding of different or rotated configurations of the synergy space is plausible in the motor system: synergies are flexible configurations (Turvey, 2007) and are therefore likely to modify or adapt to task demands rather than being rigid postural schemes (Latash et al., 2007). In addition, the solution to many distinct problems of motor control, including the Degrees of Freedom problem, is hardly unique (Bernstein, 1967). Hence, both synergies and their rotated versions may coexist in the brain, representing strategies adopted to solve the Degrees of Freedom problem in multiple possible ways, achieving the one-to-many organization that has been posited by some authors (Latash et al., 2007).

– Though the principal components obtained from the synergy space were encoded in each subject separately, we showed a highly consistent similarity of the first three synergies across subjects (Video 1). However, with the current procedure, slight inter-subject differences in the principal components could not be considered during the assessment of the topographical mapping. We believe that the topographical assessment requires the definition of stable population-level synergies to allow for the identification of the optimal rotation of each component and to test their arrangements in an independent group of subjects.

– The present study focused only on hand-grasping movements. We expect that the addition of more action types, such as non-grasping object-directed movements or intransitive actions, may lead to the description of synergies that are different from those described here for grasping. For this reason, the cortical organization described in this study may reflect only in part the overall cortical organization of hand movement control.

In summary, in this paper we reported that the three synergies topographically arranged on the cortex are the best descriptors (as Principal Components) of posture space and correspond to meaningful grasping primitives. While certainly this may not be true in the case of the rotated versions of those synergies, theoretically they could fit as well, or even better, the brain activity patterns.

We have discussed these important aspects, the caution they warrant in the generalization of our results, the requirements for additional distinct studies with specific goals in the “Limitations” and in the “Beyond synergies” sections. Finally, in agreement with the suggestions by the Reviewer, we attenuated the description of the topographical organization of hand synergies and removed the sentence stating the invariance of individual R^2^ coefficients to rotation.